# EB-gMCR: Energy-Based Generative Modeling for Signal Unmixing and Multivariate Curve Resolution

## Abstract

Signal unmixing analysis decomposes data into basic patterns and is widely applied in chemical and biological research. Multivariate curve resolution (MCR), a branch of signal unmixing, separates mixed signals into components (base patterns) and their concentrations (intensity), playing a key role in understanding composition. Classical MCR is typically framed as matrix factorization (MF) and requires a user-specified number of components, usually unknown in real data. Once data or component number increases, the scalability of these MCR approaches face significant challenges. This study reformulates MCR as a data generative process (gMCR), and introduces an Energy-Based solver, EB-gMCR, that automatically discovers the smallest component set and their concentrations for reconstructing the mixed signals faithfully. On synthetic benchmarks with up to 256 components, EB-gMCR attains high reconstruction fidelity and recovers the component number within 5% at 20dB noise and near-exact at 30dB. On two public spectral datasets, it identifies the correct component number and improves component separation over MF-based MCR approaches (non-negative variants [NMFs], ICA, MCR-ALS). EB-gMCR is a general solver for fixed-pattern signal unmixing (components remain invariant across mixtures). Domain priors (non-negativity, nonlinear mixing) enter as plug-in modules, enabling adaptation to new instruments or domains without altering the core selection learning step.

## 1 Introduction

Multivariate curve resolution (MCR) is a signal unmixing technique commonly applied to chemical data, such as optical or mass spectra (de Juan & Tauler, 2021). The core concept involves treating chemical data ($D \in \mathbb{R}^{M \times d}$, $M$ samples) as a mixture of $N$ components, where $d$ is the feature dimension (e.g., wavelengths in spectra). One primary goal of MCR is to obtain the component patterns ($S \in \mathbb{R}^{N \times d}$) and their concentrations ($C \in \mathbb{R}^{M \times N}$, Eq. 1). Usually, the formulation of MCR includes a noise term ($\varepsilon$) matching the data dimensions to represent signals introduced by chemical instruments. If a separated pattern closely approximates an actual source chemical pattern, it is possible to identify the chemical composition of the sample. Although MCR is conceptually intuitive, various practical difficulties arise in real-world applications. For instance, the types and number of mixed components are often unknown. The solution space becomes infinite in MCR unless additional constraints are imposed.

$$D = CS + \varepsilon \tag{1}$$

MCR solvers are commonly divided into two categories: iterative and non-iterative. Non-iterative solvers attempt to recover the underlying components by imposing constraints that ensure a full-rank system, then solving it with MF techniques. Typical constraints, such as orthogonality of the pattern matrix, are chosen based on the chemical properties of the data. However, real-world applications often fail to yield solvable systems, making iterative solvers a more general option. These methods optimize an objective function directly, as in Parallel Factor Analysis 2 (PARAFAC2), MCR-alternating least squares (ALS), etc (Tauler et al., 1995; Harshman, 1972). Despite their broad use, both solvers suffer from numerical instability, especially under high noise. Removing a subset of

rows from the data matrix can cause the estimated pattern and concentration matrices to deviate substantially from those derived from the full dataset, even when both sets originate from the same data acquisition process. Scalability is another challenge: when either the component c or dataset size grows large, performance deteriorates. For example, Barcaru et al. (2017) parsed GC–MS spectra using a large mass spectrum library, and Heinecke et al. (2021) analyzed metabolites by incorporating prior information from NMR signals—settings in which hundreds or thousands of candidate components may need to be considered.

A further limitation is the difficulty of adapting existing solvers to new chemical priors. Even small modifications, such as enforcing non-negativity, often require redesigning the solver because the objective function and constraints are tightly coupled. These limitations motivate a more flexible formulation of the MCR problem—one that can scale to large candidate sets, remain stable under noise, and incorporate chemical priors without redesigning the solver. To this end, we introduce generative MCR (gMCR) as a general data-generating formulation, and—built on top of it—the energy-based solver EB-gMCR. Our contributions are fourfold:

- **Dynamic component discovery.** An energy-based solver that automatically determines the minimal number of active components from a large candidate pool (tested up to 256), using the EB-select gate to identify which components participate in each mixture without manual component number ($N$) specification.

- **Generative modeling of MCR and corresponding inverse problem solver.** We reformulate multivariate curve resolution (signal unmixing) as an inverse problem on a generative process (gMCR); EB-gMCR learns a reusable solver that eliminates re-computation when analyzing new samples from the same generative process.

- **Scalable to large candidate libraries.** The selection energy optimization framework enable operation on oversize candidate pools, while the solver learns to activate only the minimal subset needed.

- **Flexible incorporation of domain priors.** The framework accommodates domain-specific constraints as plug-in modules, allowing users to tailor the solver to new problem domains without redesigning the core optimization algorithm.

We validate on synthetic mixtures and two public light spectrum datasets. The synthetic benchmark follows the standard non-negative MCR setting and allows precise scaling of unmixing difficulty; the real datasets demonstrate practical deployment. EB-gMCR achieves accurate reconstructions with a parsimonious component usage and remains competitive decomposability with MCR baselines across synthetic and real data. Although demonstrated on chemical spectra mixtures, the solver design is generic and applies to signal unmixing problems where components (base patterns) are assumed fixed.

## 2 RELATED WORKS

A wide range of approaches have been developed for MCR and signal unmixing. We briefly review the most relevant lines of work and position EB-gMCR in relation to them.

**Classical MCR Solvers.** Non-iterative methods decompose components by imposing constraints that enable direct MF, such as principal component–based approaches including self-modeling MCR and evolving factor analysis (Lawton & Sylvestre, 1971; Keller & Massart, 1991). However, adding domain-specific constraints to direct solvers is difficult, which makes iterative methods more practical for real-world chemical signals. Widely used examples include NMF and its variants such as sparse NMF and Bayesian NMF (Lee & Seung, 1999; Hoyer, 2004; Schmidt et al., 2009). Independent component analysis (ICA), nonlinear ICA, and MCR-ALS are also common approaches for unmixing chemical signals (Comon, 1994; Hyvärinen & Pajunen, 1999).

**Learning-based Unmixing.** From a machine learning (ML) view, unmixing can be posed as representation learning. Sparse autoencoders and their stricter k-sparse variants encourage compact activations (Ng, 2011; Makhzani & Frey, 2013), yet soft gates (e.g., tanh) rarely hit exact zeros; small nonzeros bleed into reconstructions and prevent truly discrete component selection. Dictionary learning learns sparse bases, but the dictionary can grow without effective control (Olshausen

& Field, 1997). Sparse unmixing formulates regression against a known dictionary and targets a minimal active set under high-fidelity reconstruction, but it requires prior libraries and can become inefficient as base dimensionality and library size grow (Xu, 2024; Shu et al., 2025). VQ-VAE trains an encoder–decoder with a discrete codebook via stop-gradient (van den Oord & Vinyals, 2017); the codebook serves compression and generation, yet latent codes do not correspond to physical components or concentrations.

**Positioning.** Under the linear mixing model $D \approx CS$: (i) fixing $S$ to a known library and optimizing $C$ yields sparse unmixing/sparse coding; (ii) fixing $C$ (e.g., to prescribed sparse codes) and optimizing $S$ yields dictionary learning/update. EB-gMCR couples selection and basis learning to learn a minimal set that achieves high-fidelity reconstructions, in contrast to VQ-VAE designs where quantized latent vectors do not expose explicit components or concentrations.

## 3 GENERATIVE MULTIVARIATE CURVE RESOLUTION (GMCR)

The classical MCR formulation represents mixed signals (data) as a linear combination of unknown $N$ components, where $N \in \mathbb{N}$. In principle, $N$ should be set large enough to ensure the true solution space is captured, especially when no prior assumptions restrict the candidate set of possible chemical components. For example, library-based searches can easily involve thousands of candidates. Specifying such a large number of components is both numerically unstable and prone to overfitting noise, which explains why most MCR approaches restrict $N$ to a small number. However, the original spirit of MCR: simultaneously identifying the known and unknown-patterns arguably essentially needs a large candidate set. To address this challenge, the gMCR framework generalizes the classical linear mixing via an aggregation function $\Phi$, reformulating the MCR equation as a generative process that describes how chemical patterns mix, offering a generative view of the MCR problem (Eq. 2; Fig. 1a). Here, $\omega$ indexes a sample from the generative process, $\delta(\omega)$ selects which candidates are active for that sample, and $\Phi$ aggregates them (e.g., summation for linear mixing). Because each component's indicator is modeled independently, $\Phi$ operates on the union of selected component sets (Eq. 3). The noise term $\varepsilon(\omega)$ represents measurement error in the observation process, independent of component selection. When $\Phi$ is linear summation (as in classical MCR), the incremental property simplifies to $\mathbf{D}_{N+1}(\omega) \approx \mathbf{D}_N(\omega) + \mathbf{D}_1(\omega)$, where $\mathbf{D}_1$ denotes the contribution of the newly added component.

$$\mathbf{D}_N(\omega) = \Phi\left(\{\delta_i(\omega), \mathbf{C}_i(\omega), \mathbf{S}_i(\omega)\}_{i=1}^N\right) + \varepsilon(\omega) \tag{2}$$

$$\mathbf{D}_{N+1}(\omega) = \Phi\left(\{\delta_i(\omega), \mathbf{C}_i(\omega), \mathbf{S}_i(\omega)\}_{i=1}^N \cup \underbrace{\{\delta_{N+1}(\omega), \mathbf{C}_{N+1}(\omega), \mathbf{S}_{N+1}(\omega)\}}_{\mathbf{D}_1}\right) + \varepsilon(\omega) \tag{3}$$

The gMCR framework enables flexible incorporation of prior knowledge by modeling the sampling function (paths), and known physical descriptions of data generation can be directly embedded into the function. **A key advantage is reusability: once trained on representative mixtures, the learned generative model can rapidly decompose new observations without retraining, as long as the underlying mixture mechanism remains unchanged.** Classical MCR becomes an inverse problem on the gMCR generative process: inferring which components and concentrations generated the observed mixtures. A fundamental difference from MF-based MCR is that a well-trained gMCR model can be reused for any new samples from the same generative process, eliminating re-optimization. This formulation also removes MF computational constraints (e.g., orthogonality, full-rank conditions) that real superposition signals may not satisfy. In this setting, data size no longer creates computational bottlenecks: samples are treated as i.i.d. draws from the modeled generative process, making mini-batch computation natural. This keeps gMCR scalable even when the data matrix reaches millions of rows—where traditional MCR solvers typically break down—mirroring practices in modern deep learning (DL) frameworks.

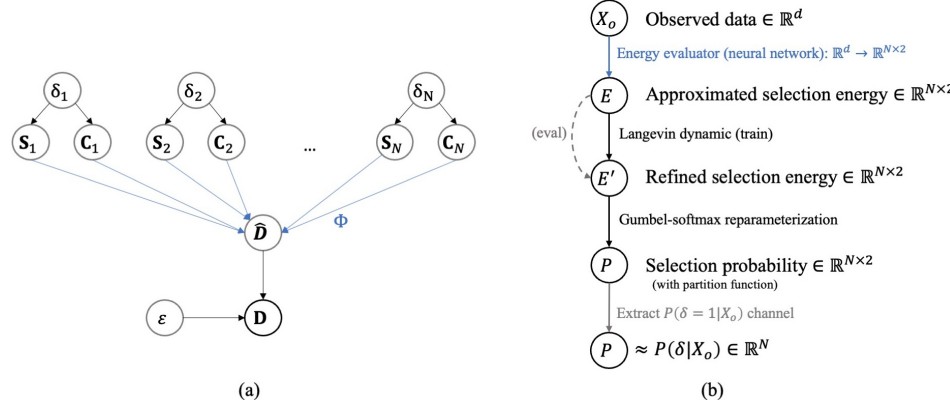

Figure 1: Overview. (a) gMCR graphical model. Only the black node $\mathbf{D}$ (data) is observed. (b) EB-select. An energy-based adaptive gate that infers the component selection ($\delta$).

## 4 ENERGY-BASED DEEP GENERATIVE MODELING FOR GMCR (EB-GMCR)

Despite the natural forward formulation of gMCR, the reality of signal unmixing is that we must solve it as an inverse problem. EB-gMCR solves this inverse problem (inferring components from observed mixtures) using energy-based modeling. Its goal is to provide a flexible signal unmixing framework that accommodates real-world mixed signals and can be readily adapted to their data characteristics. §4.1 reviews constraints derived from chemical knowledge; §4.2 introduces EB-select module for modeling the indicator function in gMCR; §4.3 details the learning algorithm; §4.4 describes termination criteria and checkpointing design, and §4.5 provides a convergence analysis of the learning algorithm.

### 4.1 CHEMICAL CONSTRAINTS INCORPORATION

In solving MCR, certain structural properties must be imposed on the concentration and component matrices to reflect prior chemical knowledge. MCR solvers commonly enforce *non-negativity of concentrations* and address the *ambiguity of components*. Depending on the measurement modality, additional properties such as *sparse components* or *non-negative components* may also be relevant. Because EBg-MCR models the entire data generation process with parametric functions, many of these can be built directly into the modeled function through data-independent mappings applied to each signal without interacting with other collected signals. For example, non-negativity can be enforced by passing the concentrations through an absolute value or a ReLU function. Remaining constraint not belonging to data-independent mapping is *ambiguity of components*, which is handled within the solver's core modules (Section 4.3).

### 4.2 ENERGY-BASED ADAPTIVE SELECTION MODULE FOR COMPONENT INDICATOR

A core distinction of gMCR is the explicit modeling of the component appearance indicator $\delta$. In over-parameterized regimes $N \gg N_{\text{true}}$, $\delta$ is expected to be sparse because only a small subset of components contributes to each mixture. **In gMCR, zeros in $\delta$ do not denote "near-zero" amplitudes; they encode non-selection (hard gating).** This induces a sparse selection over components and treats selection as a discrete event rather than a small magnitude. This viewpoint is crucial and changes learning behavior in our framework. The solver operates directly in a select / non-select environment instead of pushing continuous gate probabilities toward 0 or 1. Smooth gates (sigmoid, tanh) allow information to flow through inactive paths, creating leakage from non-selected components. Instead, EB-gMCR requires near-hard gating on component selection during both training and inference. To realize this, we introduce the energy-based adaptive selection module (EB-select; Fig. 1b), which learns sparse indicators through physics-inspired energy field optimization combined with differentiable discrete-choice relaxation. Although the intuition—inferring component presence from observed mixtures—resembles spike-and-slab sparsity (Ray & Szabó, 2022), EB-

select does not posit a sparsity prior or optimize an ELBO. Instead, it directly learns the indicator $\delta$ within an end-to-end energy optimization framework.

EB-select approximates the selection probability $P(\delta \mid X_o)$ (Eq. 4), where a neural network (NN) evaluates selection energies $E \in \mathbb{R}^{N \times 2}$ that are converted to probabilities via Gumbel-softmax reparameterization (Fig. 1b; Eq. 5). To ensure stable optimization, we regularize $E$ to maintain bounded energies during training. The energy-based formulation arises naturally from the generative framework: each observed sample $X_o$ (where subscript "o" denotes "observed," corresponding to collected data $D$ in Eq. 1) is treated as a realization from the target generative process. Under this setting, $P(X)$ is modeled as a Dirac delta $\mathrm{Dirac}(X - X_o)$, causing the marginal $P(\delta)$ to collapse to the conditional $P(\delta|X_o)$ as shown in Eq. 4. The selection energy tensor $E \in \mathbb{R}^{N \times 2}$ contains two energy values per component: one for the selected state ($\delta = 1$) and one for the non-selected state ($\delta = 0$). The Gumbel-softmax reparameterization (Jang et al., 2016; Maddison et al., 2016) converts these energies to probabilities while maintaining differentiability during training. The resulting function $f_e : \mathbb{R}^d \to [0,1]^N$ operates on individual samples $x_o \in \mathbb{R}^d$ and approximates the posterior selection probabilities (Eq. 5). During training, $f_e$ is applied independently to each observation in the dataset with temperature $\tau$ allowing gradient flow through soft probabilities; during evaluation, we set $\tau_{\text{eval}} = 0.01$ to enforce near-deterministic binary predictions.

$$P(\delta) = \int P(\delta, X)\, dX = \int P(\delta \mid X) P(X)\, dX$$

$$= \int P(\delta \mid X)\mathrm{Dirac}(X - X_o)\, dX = P(\delta \mid X_o) \tag{4}$$

$$f_e : \mathbb{R}^d \to [0,1]^N, \quad f_e(x_o) \approx \mathbb{E}[\delta \mid x_o] \tag{5}$$

To further fasten convergence, in the training mode, EB-select refines its evaluated energy using Langevin dynamics (LD), a Markov chain Monte Carlo (MCMC)-like variant suitable for improving exploration effect of gradient-based optimization (Eq. 6) In this notation, $t$ represents the step in LD, and $h$ is the scale of LD noise. The stochastic gradient Langevin dynamic (SGLD) formula is rewritten in an approximate form ( (Eq. 7) (Welling & Teh, 2011) to enable parallel computation and prevent nested loops, where $T$ is the total number of SGLD samples. To remain consistent with classical EB modeling, the selection energy must be bounded and kept it near a low-energy state. Accordingly, the overall loss includes $\lambda_{se}|E|_2^2$ to encourage the energy state toward minimal-energy configurations, similar to how physical systems naturally seek low-energy states. This term regularizes the model's output energy rather than model parameters, maintaining bounded energy scales while the Gumbel-softmax gate learns component selections.

$$e_{t+1}^{(i)} = e_t^{(i)} - \frac{h}{2}\nabla_e E_{f_e}(e_t^{(i)}) + \sqrt{h}\,\eta_t^{(i)}, \quad i = 1, \ldots, T \tag{6}$$

$$e' = e - \frac{Th}{2}\nabla_e E_{f_e}(e) + \sqrt{Th}\,\eta \tag{7}$$

## 4.3 EB-GMCR ARCHITECTURE AND OPTIMIZATION

Aside from modeling the indicator function, the sample functions for components $f_s$ and for their concentrations $f_c$ must still be constructed. To align with classical MCR (Eq. 1), the default version of $f_s$ is defined as a threshold-based retrieval from a set of learnable component vectors $\mathbf{s}_i \in \mathbb{R}^d$ (Eq. 8), where each $\delta_i$ indicates whether the $i$-th component's predicted probability $f_e(X_o)_i$ exceeds threshold $\tau_{thres}$. We set $\tau_{\text{thres}} = 0.9999994$, corresponding to a $5\sigma$ confidence level commonly used in physics for establishing existence (equivalent to $p < 3 \times 10^{-7}$). The default version of $f_c :$ $X_o \to \mathbb{R}^N$ is a vector-valued function, parameterized by a NN, that directly evaluates concentrations of components from the observed data. In practice, users can extend these two sample functions by incorporating chemical knowledge. Finally, the data generation process modeled in EB-gMCR reproduces the assumptions of the gMCR problem (Eq. 2).

$$f_s(f_e(X_o)) = \{\mathbf{s}_i \mid \delta_i = 1\}, \quad \text{where } \delta_i = \mathbb{I}\big(f_e(X_o)_i \geq \tau_{thres}\big) \tag{8}$$

With explicit access to $f_s$ and its codomain, the crucial property of "component ambiguity" can be incorporated into the EB-gMCR's learning objective. Because the EB-gMCR solver is intended for over-parameterized conditions, duplicate components are not allowed in the final solution. Inspired by kernel-based measures for feature independence (e.g., the Hilbert-Schmidt independence criterion (HSIC) (Gretton et al., 2005) the dissimilarity among stored components can be evaluated in a reproducing kernel Hilbert space (RKHS). Specifically, the off-diagonal entries of the kernel matrix, generated by pairwise evaluation of the components, serve as a penalty (regularization term) weighted by hyperparameter $\lambda_{amb}$ to promote dissimilar components (Eq. 9). In this formulation, $K$ is unspecified but must fulfill the definition of a valid kernel function (with the semi-positive definite property). The default choice in the EB-gMCR is the radial basis function (RBF) kernel.

$$\mathcal{R}(X_o) = \sum_{\substack{j,k \in \mathcal{I}_{X_o} \\ j < k}} K(\mathbf{s}_j, \mathbf{s}_k), \quad \text{where } \mathcal{I}_{X_o} = \{i \mid f_e(X)_i \geq \tau_{thres}\} \tag{9}$$

Although the HSIC regularizer mitigates duplicate patterns, heavy over-parameterization introduces a second risk: a single ground-truth component can be represented by a linear combination of several learned patterns, thereby inflating the effective component number. To steer EB-gMCR toward the smallest feasible set, the solver introduces a component usage energy term $C(X_o)$ that explicitly encourages minimal component usage (Eq. 10). Unlike standard L1 sparsity penalties that regularize model parameters, $C(X_o)$ operates on the energy state itself: it measures the average selection probability across all $N$ candidates, normalized by pool size for numerical stability. Because the Gumbel-softmax gate operates identically during training and evaluation (only temperature $\tau$ differs), as $\tau \to 0$, $C(X_o)$ approaches the discrete component number divided by $N$, making it an effective soft-to-hard approximation. When weighted by the Lagrange multiplier $\lambda$, this term creates an energy landscape favoring parsimonious solutions. The multiplier $\lambda$ is activated dynamically once reconstruction quality (measured by mean squere error [MSE]) falls below a preset threshold, shifting the optimization from "reconstruct accurately" to "reconstruct with minimal components."

$$C(X_o) = \frac{1}{N} \sum_{i=1}^{N} f_e(X_o)_i \tag{10}$$

Finally, by explicitly defining all parametric functions in the EB-gMCR solver, one can approximate gMCR by enforcing that generated data ($X_g$) remain close to the true data ($X_o$). Under the Gaussian noise assumption for $\varepsilon$ (Eq. 11, the negative log-likelihood of $X_o$ given the model parameters is proportional to the MSE (Eq. 12; LeCun et al. (2006)). Since the Gaussian likelihood is already normalized in closed form, no explicit partition function or score matching is required. Therefore, EB-gMCR simply minimizes the total energy (Eq. 13), combining MSE reconstruction, component usage $C(X_o)$, selection energy $\|E\|_2^2$, and ambiguity regularization $\mathcal{R}(X_o)$.

$$X_g = \Phi(f_e, f_s, f_c) + \varepsilon, \quad \varepsilon \sim \mathcal{N}(0, \sigma^2 I) \tag{11}$$

$$-\log p(X_o \mid f_e, f_s, f_c) \propto \|X_o - X_g\|^2 \tag{12}$$

$$\mathcal{L}_{\text{EB-gMCR}} = \|X_o - X_g\|^2 + \lambda \cdot C(X_o) + \lambda_{\text{se}}\|E\|_2^2 + \lambda_{\text{amb}}\mathcal{R}(X_o) \tag{13}$$

### 4.4 CHECKPOINTING SOLVER

EB-gMCR reverses the traditional MCR workflows: it launches from an intentionally oversized pool ($N \gg 10$) and prunes away redundant components. The EB-select regularizer penalizes idle components, steering the solver to reproduce the observed data $X_o$ with the fewest active components. Synthetic trials show a characteristic trajectory: selection energy and the number of active

components drop early; as energy keeps falling, the active set grows again and may oscillate until the energy minimum. Reconstruction degrades only when the active count dips below the true rank, then recovers as components reactivate—often matching final accuracy well before the minimum. Therefore, EB-gMCR solver exploits this behavior through **minimum component checkpointing**:

- **User-defined target band.** A reconstruction is considered eligible when its coefficient of determination falls inside the user-specified band (e.g., $0.975 \leq R^2 \leq 0.980$).
- **Single rolling checkpoint.** The checkpoint in target bands is replaced only if the new generative function uses fewer components, ensuring the most parsimonious solver.
- **Termination.** Optimization stops once the EB-select energy drops below the threshold $\|E\|_2^2 < \sigma \|E_{init.}\|_2^2$, with $\sigma = 0.25$ — a value lower enough to entering oscillatory phase based on preliminary test.

The full optimization loop is summarized in Algorithm 1, and additional training details are provided in Appendix A. EB-gMCR adopts the following module defaults: $f_s$ is a learnable set of component vectors, $f_e$ is a 3-layer NN with tanh activations, and $f_c$ is a 3-layer NN with ReLU activations predicting concentrations. Under this definition, $R^2$ can be less than -1 when residual errors greatly exceed the variance of the observed data.

## 4.5 Convergence Analysis

**What We Prove: Algorithmic Convergence.** The learning dynamics of EB-gMCR exhibit a two-phase structure (Corollary B.1), enabling an end-to-end training paradigm:

- **Phase A (support discovery):** Once the component usage penalty becomes sufficiently active ($\lambda \geq \lambda^*$ in Eq. 13), per-component energy gaps concentrate uniformly with deviation $O(\sqrt{\log(N|W|)/M})$ over the window. The selection gates stabilize after horizon $T_{\text{supp}}(\alpha)$ that decreases with dataset size $M$.
- **Phase B (optimization on fixed support):** Once gates stabilize, the component usage becomes constant. The reduced objective is $L$-smooth and satisfies the Polyak-Łojasiewicz (PL) condition, yielding linear convergence to a variance floor (Theorem B.6).

The $M$-dependence in Phase A predicts faster convergence with larger datasets, consistent with empirical observations: $8N$-sample runs converge in approximately 2 hours vs 3.3 hours for $4N$-samples (Table 1). Convergence is measured in mini-batch iterations rather than full data passes over $X_o$.

**What We Don't Prove: Solution Quality.** Our analysis establishes *convergence to an energy minimum* but does not prove that this minimum yields the correct component number or decomposition. The theory characterizes the optimization dynamics but not whether the converged solution matches the true mixture composition. Empirical validation (§5.1) establishes that the energy landscape design yields accurate decompositions in practice.

## 5 Experimental Results and Discussions

**Evaluation Metrics.** We evaluate decomposition quality using $R^2$ (coefficient of determination), the standard metric for linear mixture reconstruction in MCR. $R^2 = 1 - SS_{\text{res}}/SS_{\text{tot}}$ measures how well the linearly reconstructed signal $\hat{X} = CS$ matches the observed data $X_o$; values near one indicate accurate linear factorization. EB-gMCR additionally reports Estimated Component Number (EC), counting components with selection probability exceeding $\tau_{\text{thres}} = 0.9999994$ ($5\sigma$ threshold). Baseline methods do not report EC as they require $N$ as a fixed input.

## 5.1 Synthesized Mixed Component Dataset

**Dataset Generation.** Real spectra seldom reveal pure constituents or concentrations, so we use an in-silico benchmark (Appendix C). Components are nonnegative, unit-norm vectors in $\mathbb{R}^{512}$. For each mixture, the active component number is drawn uniformly from $\{1, 2, 3, 4\}$, and concentrations

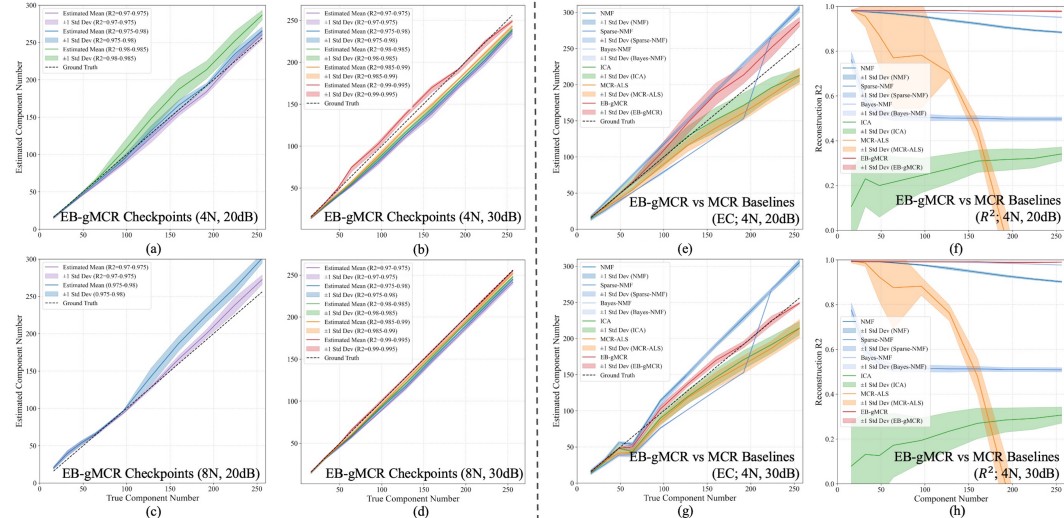

Figure 2: Synthetic benchmarks. (a–d) EB-gMCR checkpoints: estimated vs. true component number (dashed black line); mean (solid) and ±1 SD (shaded) over 5 replicates; colors denote $R^2$ checkpoint bands. Panels: (a) $4N$, 20dB; (b) $4N$, 30dB; (c) $8N$, 20dB; (d) $8N$, 30dB (e, g) EB-gMCR vs. baselines: estimated vs. true components at 4N under 20 dB and 30 dB. (f, h) Reconstruction $R^2$ at each method's EC for the same settings.

from $[1, 10]$. We evaluate $N \in \{16, ..., 256\}$ with dataset size $4N$ and $8N$ (64-2048 mixtures). Each configuration is run 5× for EB-gMCR and 100× for baselines. Since EB-gMCR learns the rank from data while baselines require a chosen rank, we report a common trade-off: high $R^2$ with few active components; search ranges, stopping rules, and other settings are in Appendix D.

**Component Estimation of Checkpoints.** Figs. 2 (a)-(d) summarize EB-gMCR on synthetic benchmarks across sample size ($4N$, $8N$) and noise (20dB and 30dB). Results are reported by reconstruction $R^2$ checkpoints; each curve shows mean estimated count with ±1 SD. The trend is consistent: higher noise or fewer samples activates more components, whereas lower noise and larger samples recover near-true component numbers across 16–256. Checkpoints offer a clear fidelity–parsimony dial: higher $R^2$ favors fidelity with a small number usage bias, lower $R^2$ favors parsimony.

**Benchmarking Against MCR Baselines.** Using the best EB-gMCR checkpoints (see Appendix D; $4N$, 20dB: $R^2$ 0.98–0.985; $4N$, 30dB: 0.99–0.995) we compare against NMF, sparse-NMF, Bayes-NMF, ICA, and MCR-ALS (Fig. 2e–h). Sparse-NMF, ICA, and MCR-ALS degrade beyond 32–64 components and become unreliable above 100; they remain competitive at 16 components (Table 3) with lower compute. NMF and Bayes-NMF sustain high $R^2$ at larger scales but do so by overestimating component number. EB-gMCR maintains high $R^2$ with EC near the ground truth across the range, yielding the strongest decomposability (fidelity–parsimony trade-off).

**Computational Efficiency (Table 1):** Traditional MCR methods face a fundamental search density dilemma—sparse intervals (e.g., 32, 64, 96, ...) are fast but risk missing the optimal component count, while dense search (interval=1 or 2) ensures reliability but becomes computationally prohibitive at scale. We measured baseline wall-time at intervals of 32, 16, and 8 searching from interval→256; even at interval=8, baselines require 0.02–5.15 hours cumulative. Extrapolating to interval=1 (required for reliable component discovery near N=256) would increase costs by ∼3–5× (0.1–26 hours). EB-gMCR eliminates this dilemma through energy-based selection: a single training run (2.1–3.3 hours) automatically identifies optimal component count without exhaustive search. Moreover, baselines must re-optimize for each new dataset, while EB-gMCR's trained model enables fast inference on new samples. Overall, EB-gMCR extends MCR to the hundred-component regime with principled automatic component discovery.

Table 1: Wall-clock time: component search at interval=$\{8, 16, 32\}$(interval $\rightarrow$ 256; $N_{\text{true}} = 256$ SNR=20dB) on Intel i9-14900K and Nvidia RTX A6000 (only EB-gMCR use GPU).

| Search Interval | NMF | Sparse-NMF | Bayes-NMF | ICA | MCR-ALS | EB-gMCR |
|---|---|---|---|---|---|---|
| | | | *4N (1024 samples)* | | | |
| 32 | 0.029±0.000 | 0.009±0.000 | 0.073±0.001 | 1.06±0.01 | 0.394±0.031 | |
| 16 | 0.055±0.001 | 0.017±0.000 | 0.139±0.002 | 1.75±0.00 | 0.751±0.029 | 3.30±0.60 |
| 8 | 0.104±0.002 | 0.032±0.001 | 0.268±0.006 | 3.16±0.01 | 1.41±0.03 | |
| | | | *8N (2048 samples)* | | | |
| 32 | 0.006±0.000 | 0.009±0.000 | 0.016±0.001 | 1.49±0.00 | 0.484±0.031 | |
| 16 | 0.012±0.000 | 0.017±0.000 | 0.030±0.001 | 2.71±0.01 | 0.893±0.003 | 2.10±0.20 |
| 8 | 0.023±0.000 | 0.032±0.000 | 0.056±0.002 | 5.15±0.02 | 1.81±0.05 | |

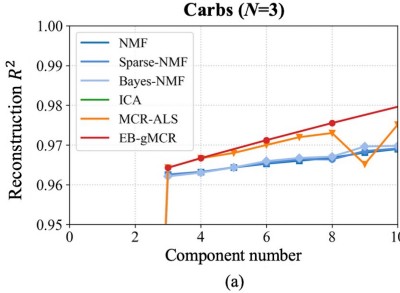 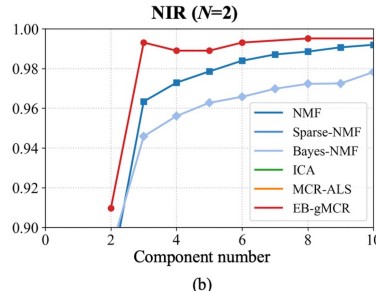

(a)  (b)

Figure 3: Real-data reconstruction. $R^2$ vs component number: (a) Carbs ($N = 3$), (b) NIR ($N = 2$).

## 5.2 REAL-WORLD CHEMICAL DATASET

**Datasets.** To assess decomposability on real chemical signals, we use **Carbs** (Raman carbohydrates), and **NIR** (first-derivative NIR) (Engelsen, n.d.; Liebmann et al., 2009). Carbs: $M = 22$ mixtures in $\mathbb{R}^{1401}$ from $N = 3$ non-negative sources; concentrations lies in $[0, 1]$; linear mixing with noise. NIR: $M = 166$ mixtures in $\mathbb{R}^{235}$; signals include negatives; two-analyte concentrations are known while components are unknown, so we report reconstruction $R^2$ and concentration error. EB-gMCR handles sign by removing the absolute-value step in the forward pass. For baselines constrained to non-negativity, we use a sign-splitting transform (separate ± parts, flip the negative part's sign, concatenate to a nonnegative $2d$ vector, fit, then invert). Baselines follow the adaptive rank sweep of Smith et al. (2019), increasing $N$ from 1 until $R^2$ stabilizes. For EB-gMCR we initialize 128 candidate components (since $M$ bounds a meaningful basis) and stop when selection energy and the active set stabilize.

**Decomposability.** EB-gMCR converges to $N = 3$ on **Carbs** and $N = 2$ on **NIR**, matching ground truth, and achieves the top reconstruction $R^2$ at those counts (Fig. 3). Checkpoints are saved by $R^2$ bands, so plots show saved states rather than every EC. To assess decomposability, we align recovered components and concentrations via linear assignment, reporting cosine similarity and RMSE for $S$ and RMSE for $C$. On Carbs, all methods except ICA show similar decomposability, with EB-gMCR on par with MCR-ALS; on NIR, EB-gMCR attains the best reconstruction and the best decomposability at the true rank. Full metric values evaluated at ground truth component number appear in Table 5. Although EB-gMCR can take longer wall time in the $N < 10$ regime, it does not sacrifice accuracy—results are comparable on the classical Carbs set and strongest on the non-classical NIR dataset.

## 5.3 ABLATION STUDY

**Ablations.** We start from the standard EB-gMCR (EB-select gate, MSE loss + SGLD exploration + count penalty $C(X_o)$ + minimum selection-energy term $\|E\|_2^2$ + HSIC redundancy $\mathcal{R}(X_o)$, with checkpointing) and remove one loss component at a time: w/o $C(X_o)$, w/o $\|E_{f_e}\|_2^2$, and w/o $\mathcal{R}(X_o)$. We rerun the 256-component case at 20/30 dB and $4N/8N$. Table 2 reports reconstruction $R^2$, EC, and wall time. A fixed-rank variant is omitted since 1024 unconstrained components reduces to an NMF-like fit and does not probe selection.

Table 2: Ablation study for EB-gMCR ($N = 256$) on Nvidia RTX A6000.

| Solver | Noise=20dB | | | Noise=30dB | | |
|---|---|---|---|---|---|---|
| | $R^2$ | EC | Time (hr.) | $R^2$ | EC | Time (hr.) |
| **Data size = 4N (1024 mixed samples)** | | | | | | |
| EB-gMCR | 0.978±0.003 | 278.0±13.4 | 3.6±0.8 | 0.990±0.000 | 248.8±1.1 | 3.3±0.6 |
| w/o SGLD | 0.980±0.001 | 292.0±8.2 | 4.7±0.3 | 0.990±0.000 | 257.8±6.2 | 2.9±0.3 |
| w/o $\mathcal{C}(X_o)$ | 0.980±0.001 | 284.6±13.4 | 4.1±0.3 | 0.995±0.000 | 268.2±4.5 | 4.1±0.3 |
| w/o $\|E\|_2^2$ | 0.979±0.002 | 282.0±20.3 | 6.1±0.2 | 0.990±0.000 | 251.8±3.4 | 6.0±0.4 |
| w/o $\mathcal{R}(X_o)$ | 0.980±0.000 | 292.2±13.0 | 3.8±0.6 | 0.990±0.000 | 249.0±1.6 | 4.5±0.7 |
| **Data size = 8N (2048 mixed samples)** | | | | | | |
| EB-gMCR | 0.975±0.000 | 301.0±6.4 | 2.0±0.2 | 0.991±0.001 | 255.0±1.4 | 2.1±0.2 |
| w/o SGLD | 0.976±0.002 | 323.8±41.2 | 2.5±0.3 | 0.991±0.000 | 255.8±0.4 | 2.0±0.3 |
| w/o $\mathcal{C}(X_o)$ | 0.975±0.000 | 318.6±32.2 | 2.2±0.1 | 0.993±0.002 | 298.2±29.3 | 2.4±0.2 |
| w/o $\|E\|_2^2$ | 0.975±0.000 | 290.4±11.9 | 12.1±0.1 | 0.979±0.000 | 255.6±0.5 | 12.2±0.1 |
| w/o $\mathcal{R}(X_o)$ | 0.975±0.002 | 305.4±6.4 | 2.5±0.5 | 0.991±0.000 | 254.8±0.8 | 2.2±0.1 |

**Findings.** The MSE term already yields high $R^2$; auxiliaries mainly improve sparsity and efficiency. Removing SGLD is acceptable at 30 dB but inflates EC and runtime at 20 dB. Dropping $\mathcal{C}(X_o)$ barely changes $R^2$ yet increases EC (over-use). Removing $\|E_{f_e}\|_2^2$ prevents timely convergence and lengthens wall time. HSIC has little effect on these near-orthogonal synthetic sources. Overall, the full configuration is the only setting that keeps EC near truth and runtime low across noise and sample regimes.

## 5.4 LIMITATION AND POTENTIAL OF EB-GMCR

Most experiments follow the classical MCR setting; at modest data sizes and component numbers, traditional solvers remain competitive in both time and decomposability. EB-gMCR incurs higher wall-clock time because it relies on trial-and-error signal reconstruction to approximate the generation process without prior knowledge (e.g., rough component number or mathematical properties of concentrations and components). Despite this computational overhead, the trained model can be reused and run inference on any new sample. Also, We do not study the ill-posed case $N > M$; in that regime the method may underestimate $N$. Sufficient mixed-data collection is an important prerequisite for applying EB-gMCR to signal unmixing.

Beyond reusability, EB-gMCR scales advantageously as both data size and component count increase. In mass spectral analysis, thousands of library patterns can be fixed as known components, allowing the network to learn only the selection gate and concentrations. Moreover, solver accuracy improves with more spectra, transforming a data-volume bottleneck into an advantage. The framework is easily extensible: physical constraints (linear or nonlinear) integrate into the forward model. Replacing a linear mixing rule with an exponential one requires only a forward-pass change, since the loss depends on selection energies and the RKHS duplication penalty, not on the specific mixing law or component representation.

## 6 CONCLUSION

We reformulate MCR as a generative process (gMCR) and solve it via energy-based modeling (EB-gMCR), which automatically discovers component counts without manual search. Synthetic experiments validate scalability with accurate component recovery up to $N$=256 components, while real datasets (**Carbs**, **NIR**) validate correctness with proper identification and competitive reconstruction. The framework extends fixed-pattern unmixing to regimes ($N > 100$) where traditional methods require exhaustive search or fail, enabling high-throughput spectral analysis and offering a transferable approach for decomposing complex superposition signals.

CHECKLIST OF REQUIRED STATEMENTS

In accordance with the ICLR 2026 author guidelines, we include the following: **Author Contributions**, **Reproducibility Statement**, and **Ethics Statement** at the beginning of the Appendix.

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

# APPENDIX

## AUTHOR CONTRIBUTIONS

All conceptual development, experimental design, and implementation were carried out by the first author. The second author is listed per lab authorship policy.

## REPRODUCIBILITY STATEMENT

We release the full EB-gMCR solver, experiment runners, and summarized artifacts as part of the submission. The code bundle (**ebgmcr/**) includes training and evaluation modules for both synthetic and real datasets, together with scripts for the main benchmarks (§ 5.1), ablation study (§ 5.3), and real-data experiments (§ 5.2). To keep the package compact, we exclude large model weights and instead provide lightweight summaries (**\*.csv**, **\*.xlsx**) that allow all figures (Figs. 2–3) and tables (Tables 2–5) to be regenerated directly via the included Jupyter notebooks. Full experiments can be rerun using the provided **\*.sh** and **\*.py** , which reproduce the raw results in raw_results. Environment details and deterministic settings are specified in the README.txt. This setup ensures that reviewers can (i) verify figures and tables without heavy compute and (ii) rerun the solver end-to-end if desired. We also include new files, including the program and result, which reproduce the Table 1 in the renewed attachment.

## ETHICS STATEMENT

This work develops a general solver for signal unmixing and is evaluated on synthetic mixtures and two public datasets: the SPECARB Raman carbohydrate database and the NIR bioethanol dataset. No human subjects, animal data, or personally identifiable information were used. The datasets are publicly available under their stated licenses, and we cite the original sources. The methods are intended for scientific data analysis in chemistry and related fields; we are not aware of specific dual-use or misuse risks beyond those inherent in generic ML research. Experiments were run on server with Intel i9 14900K CPU and NVIDIA RTX A6000 GPUs, and we report hardware/time settings to allow assessment of environmental cost.

Large language models (ChatGPT and Claude) were used as an auxiliary tool. Their role was limited to generating draft code for evaluation routines, baseline implementations, and plotting utilities; assisting with literature searches for related work; and providing editorial suggestions for the convergence proof and for language polish. All outputs were verified by the authors. All theoretical ideas, algorithmic designs, and the main framework were conceived independently by the authors.

# A  REPRODUCIBILITY DETAILS OF EB-GMCR SOLVER

## A.1  TRAINING SCHEDULES

- **Optimizer.** AdamW (learning rate $5 \times 10^{-4}$, $\beta_1 = 0.9$, $\beta_2 = 0.995$, weight decay $10^{-3}$), batch size 64, up to $T$ epochs.
- **Temperature (EB-select).** Start $\tau_0 = 1$; decay every 100 epochs: $\tau_{t+1} = \max(\tau_{\min}, \gamma \tau_t)$ with $\tau_{\min} = 0.4$ and $\gamma = 0.999$.
- **Evaluation temperature.** $\tau_{\text{eval}} = 0.01$ for near-binary gates.
- **Checkpointing.** Save a checkpoint each epoch using the rule in Appendix A.2.

## A.2  CHECKPOINT RULE WITH COMPACTNESS TIE-BREAK

- Goodness of fit: $R^2 = 1 - \frac{\|D_0 - \hat{D}\|_F^2}{\|D_0 - \bar{D}_0\|_F^2}$
- Bands: $[0.970, 0.975)$, $[0.975, 0.980)$, ..., $[0.995, 1.000)$.
- Also report $\text{nMSE} = \frac{\|D_0 - \hat{D}\|_F^2}{\|D_0\|_F^2}$ and the active-set proxy $C(f_e)$ from Appendix B.4.
- Replacement rule inside a band b: new model replaces the current best if, in order: (i) same $R^2$ band and smaller $C(f_e)$; (ii) higher $R^2$.

## A.3  RECONSTRUCTION-OSCILLATION DETECTOR (EARLY-STOP)

- **Observation.** On the synthetic dataset, after EB-gMCR first reaches its current best $R^2$ band, the reconstruction can keep entering and leaving that band while the total gate level stays high. This oscillation does not always appear on real datasets. We recommend using the detector as an auxiliary early-stopping signal. When minimum component usage is the primary goal, run longer.
- **Selection gate energy and ratio.** The initial energy $E_{\text{init}} \leftarrow \sum_{t=1}^{T} |f_e(D; \tau)|^t$ computed over the stage before the solver can reconstruct the observed data (the first entry into a recorded $R^2$ band). The target ratio is set to $0.25$. On the synthetic dataset, oscillation typically starts once the energy ratio clearly exceeds $0.25$ (this term also appears in the loss). In typical runs, optimization later reduces the sum of selection energy to a ratio near $0.05$.
- **Reconstruction oscillation.** After the solver can reconstruct the observed data, selection-energy minimization mainly guides the optimization endpoint rather than dominating the selection outcome. The outcome probability can remain unchanged when the positive and negative gate values are close. Hence, before the final energy-minimization stage, the solver has usually already identified suitable $f_e$, $f_s$, and $f_c$. If EB-select moves off the Pareto front during energy minimization, reconstruction performance leaves the target $R^2$ band. We refer to this behavior as **oscillation**.
- **Early stop decision.** If oscillation is detected and the selection-energy ratio falls below the target, the solver is treated as converged and the program stops early.

## A.4  LOSS, COEFFICIENTS, AND COMPONENT-COUNT PROXY

- **Reconstruction (common MCR).** $\hat{D} \leftarrow C(\delta \odot S)$ ($\odot$ denotes Hadamard product).
- **Objective.** $L \leftarrow \|X_o - X_g\|^2 + \lambda \cdot \mathcal{C}(X_o) + \lambda_{\text{se}} \|E\|_2^2 + \lambda_{\text{amb}} \mathcal{R}(X_o)$ (Eq. 13).
- **Default coefficients.** $\lambda_{se} = \lambda_{amb.} = 1 \times 10^{-10}$. $\lambda$ is a Lagrange multiplier computed by scaling the component minimization loss equal to MSE loss value.

## A.5  EB-GMCR TRAINING

This section provides the pseudo code EB-gMCR training procedure (Alg. 1).

---

**Algorithm 1** EB-gMCR Solver: Training and Checkpointing Loop

---

**Require:** Observed data $X_o$; learnable components $f_s$; EB-select $f_e$; concentration predictor $f_c$

1: **Initialize:** $\tau \leftarrow \tau_0$;  $E^\star \leftarrow 0.25 \times E_{\text{init}}$;  $\mathcal{R}_K \leftarrow 0$;  best $\leftarrow \varnothing$
2: **repeat**                                                                                  ▷ per epoch
3:   **for** minibatch $X_o$ **do**
4:       $\delta, E \leftarrow f_e(X_o; \tau)$;  $C \leftarrow f_c(X_o)$;  $S \leftarrow f_S$
5:       $\hat{X} \leftarrow C(\delta \odot S)$
6:       $L \leftarrow \text{MSE}(\hat{X}_o, X_o) + \lambda \mathcal{C}(f_e) + \lambda_{\text{se}}\|E\|^2 + \lambda_{\text{amb}} \cdot \mathcal{R}_K(X_o)$    ▷ Eq. 13
7:       Update parameters with AdamW to minimize $L$
8:       $\tau \leftarrow \max(\tau_{\min}, 0.999 \cdot \tau)$
9:   **end for**
10:  $\mathcal{M} \leftarrow \{R^2, \text{MSE}, \text{nMSE}, \mathcal{C}(f_e)\}$
11:  **if** $\mathcal{M}$ improves in current $R^2$-band **then**
12:      Save checkpoint; **best** $\leftarrow$ current model
13:  **end if**
14:  $\lambda \leftarrow 0.95\,\lambda + \dfrac{0.05}{\text{MSE} \cdot \mathcal{C}(f_e)}$
15: **until** convergence or usage oscillation detected
16: **return** best checkpoint

---

# B  CONVERGENCE AND SAMPLE-SIZE ACCELERATION OF EB-GMCR

## B.1  PROBLEM SETTING AND NOTATION

Let $X_o \in \mathbb{R}^{M \times d}$ denote $M$ i.i.d. observations, each a mixed signal. EB-gMCR maintains $N$ candidate components, and $X_\theta(\delta)$ denotes the data reconstruction produced by parameter $\theta$ and component selection $\delta$. The EB-select module $f_e$ outputs, for each candidate $j$, a two-logit energy vector $e_j(X_o; \theta) \equiv (e_j^{\text{sel}}(X_o; \theta), e_j^{\text{rej}}(X_o; \theta)) \in \mathbb{R}^2$. Selection probabilities use a 2-logit Gumbel-softmax (concrete relaxation). The training loss (empirical risk) equals

$$\hat{\mathcal{L}}(\theta, \delta) = \underbrace{\|X_\theta(\delta) - X_o\|_F^2}_{\text{reconstruction}} + \underbrace{\lambda C(X_o)}_{\text{usage}} + \lambda_{se}\underbrace{\frac{1}{N}\sum_{j=1}^N \mathbb{E}\Big[(e_j^{\text{sel}})^2 + (e_j^{\text{rej}})^2\Big]}_{\text{statewide quadratic}} + \underbrace{\lambda_{\text{amb}}R(X_o)}_{\text{smooth regularizer}}. \quad \text{(B.1)}$$

where $C(X_o)$ is the active component number implied by $\delta$ the "statewise quadratic" term is the computation implemented in the source code, and $R$ s any differentiable auxiliary regularizer (e.g., HSIC) with Lipschitz gradient. The HSIC regularizer is used for component uniqueness and plays no essential role in below convergence analysis. Define the population margin for component $j$

$$\Delta_j(\theta) := \mathbb{E}\Big[e_j^{\text{rej}}(X_o; \theta) - e_j^{\text{sel}}(X_o; \theta)\Big]. \quad \text{(B.2)}$$

We write $a \lesssim b$ if $a \le Cb$ for an absolute constant $C > 0$.

## B.2  ASSUMPTIONS

**Assumption B1 (Data model and noise).** For each $i \in \{1, \ldots, M\}$, $x_i = S^*c_i^* + \varepsilon_i$ where $\{\varepsilon_i\}$ are i.i.d., mean-zero sub-Gaussian random vectors in the sense of Vershynin (2018), Def. 3.4.1: for any unit $u \in \mathbb{S}^{d-1}$ and all $t \in \mathbb{R}$,

$$\mathbb{E}\exp(t\langle u, \varepsilon_i\rangle) \le \exp(\tfrac{1}{2}\sigma^2 t^2).$$

*Remark.* This tail condition enables the Hoeffding/Bernstein-type concentration and the union bound used later (e.g., in Lemma B.2) to control empirical energy-gap deviations; the linear mixing part is already specified in the main text, and Gaussian noise is a special case.

**Assumption B2 (Population energy gap).** With $\Delta_j(\cdot)$ defined in § B.1, there exist $\theta^*$ and $\gamma_0 > 0$ such that

$$\Delta_j(\theta^*) \geq \gamma_0 \text{ for } j \in S^*, \qquad \Delta_j(\theta^*) \leq -\gamma_0 \text{ for } j \notin S^*.$$

*Remark.* This anchors the true support; combined with the assumption B1 it yields sign-correct empirical gaps used in the margin concentration and gate-stabilization steps.

**Assumption B3 (Energy gap regularity).** With $\Delta_j(\cdot)$ defined in § B.1, there exists $L_\Delta > 0$ such that for $\theta, \theta' \in \Theta$,

$$|\Delta_j(\theta) - \Delta_j(\theta')| \leq L_\Delta \|\theta - \theta'\| \quad \text{for every } j.$$

Equivalently, the per-sample state energies have uniformly bounded parameter-Jacobians on $\Theta$, so $\|\nabla_\theta \Delta_j(\theta)\| \leq L_\Delta$ wherever the gradient exists for $\theta$ along training trajectory (contained in a compact $\Theta$).

**Assumption B4 (Gate temperature control).** The binary Gumbel-softmax gate uses a nonincreasing schedule $\{\tau_t\}$ with $0 < \tau_t \leq \tau_0$; evaluation uses $\tau_{\text{eval}} \approx 0$. In particular, the gate map is uniformly Lipschitz in margin $\Delta$ with constant $\leq 1/(4\tau_0)$.

*Remark.* This covers both the sampled form $y = \sigma((\Delta + Z)/\tau)$ with $Z \sim \text{Logistic}(0, 1)$ and the deterministic softmax; the bias toward the state follows from assumption B2 and does not depend on $\tau$.

**Assumption B5 (Exact-penalty window for component usage).** There exists a training interval $[t_1, t_2]$ with $\lambda_t \geq \lambda^*$ such that, during this window, the penalized objective (Eq. B.1) and the support constrained problem

$$\min_\theta f(\theta) \quad \text{s.t. } g(\theta) \leq s$$

share the same local minimizers and the same support support size $s$ (Classical $\ell_1$ exact-penalty equivalence; see Nocedal & Wright (2006), Theorem 17.3, applied to the single inequality $h(\theta) = g(\theta) - s \leq 0$ under LICQ/MFCQ).

*Remark.* EB-gMCR use Concrete gates, so $g(\theta) = \sum_j p_j(\theta)$ is a smooth surrogate of the $\ell_0$ count. We only require the existence of a $\lambda$-on window $[t_1, t_2]$ with $\lambda_t \geq \lambda^*$ (and bounded gate temperature per B4) long enough for support stabilization. $\lambda_t$ need not be monotone. After the support is fixed, the usage term is constant on that face, so later changes of $\lambda_t$ do not affect Phase B.

**Assumption B6 (Phase-B geometry: $L$-smooth + PL).** After support stabilizes, the reduced objective $f(\theta)$ (reconstruction + smooth regularizers; gate held fixed) has an $L$-Lipschitz gradient and satisfies the Polyak–Łojasiewicz (PL) inequality on a neighborhood of a minimizer:

$$\frac{1}{2}\|\nabla f(\theta)\|^2 \geq \mu\big(f(\theta) - f^*\big) \quad \text{for some } \mu > 0.$$

With $L$ as in the standard smoothness condition. (See Eq. (2) for $L$-smoothness and Eq. (3) for PL in Karimi et al. (2016).)

*Remark.* In Phase B the gate is fixed, so $f$ is a smooth nonlinear least-squares objective. If the residual Jacobina is full column rank near a minimizer, $f$ is locally strongly convex; strong convexity implies the PL inequality with the same $\mu$. The usage penalty is constant on the fixed support and does not affect this geometry.

**Assumption B7 (Energy-space Langevin refinement; Eqs. (6)–(7)).** During training, the two-logit energy vector $e$ in EB-select is refined by the SGLD micro-step

$$e_{t+1}^{(i)} = e_t^{(i)} - \frac{h}{2}\nabla_e E_{f_e}\big(e_t^{(i)}\big) + \sqrt{h}\,\eta_t^{(i)}, \qquad i = 1, \ldots, T,$$

where $\eta$ is independent of the mini-batch choice at that step. We equivalently use the approximated form

$$e' = e - \frac{Th}{2}\nabla_e E_{f_e}(e) + \sqrt{Th}\,\eta, \qquad \eta \sim \mathcal{N}(0, I).$$

Hence, the injected noise has covariance $hI$ per micro-step and $ThI$ for the approximated step. The fate acts on the margin $\Delta = e_{\text{rej}} - e_{\text{sel}}$; after refinement,

$$\Delta' = \Delta - \frac{Th}{2} g_\Delta + \sqrt{Th}\, \zeta,$$

where $\zeta = \eta_{\text{rej}} - \eta_{\text{sel}}$ has zero mean and variance $\leq 2$ (equals 2 if those Gaussians are independent). Only these second-moment bounds are used in the analysis.

*Remark.* Nosie is applied to energies (not parameters) and only during training; under assumption B6 it yields the usual linear convergence to a variance floor and does not change the minimizer.

### B.3 PHASE SEPARATION

**Corollary B.1 (Phase separation under an exact-penalty window).** *Assume **B1–B5**. Let $[t_1, t_2]$ be a window with $\lambda_t \geq \lambda^*$ and $\tau_t \leq \bar{\tau}$ for all $t \in [t_1, t_2]$, and suppose*

$$|[t_1, t_2]| \geq T_{supp}(\alpha/2), \qquad |[t_1, t_2]| := t_2 - t_1 + 1$$

*Then, with probability at least $1 - \alpha$, there exists $t^* \in [t_1, t_2]$ such that $S_t = S^*$   for all $t \geq t^*$.*

*We call $t < t^*$ **Phase A (support selection)** and $t \geq t^*$ **Phase B (optimization on the fixed support)**. Moreover, for $t \geq t^*$ the usage term is constant on the face $\{S_t = S^*\}$. If **B6–B7** hold for the reduced objective $f$ (gate fixed), then for any $\eta \in (0, 1/L]$ there exist $\rho_t \in (0, 1)$ and $V_t \geq 0$ such that*

$$\mathbb{E}[f(\theta_{t+1}) - f^* \mid \mathcal{F}_t] \leq \rho_t (f(\theta_t) - f^*) + V_t, \qquad (t \geq t^*).$$

*In particular, under the PL inequality (B6), one may take $\rho_t = 1 - \mu\eta_t$ and $V = \frac{L\eta\nu^2}{2\mu}$ is the variance Lemma  B.5 / Theorem  B.6.*

*Proof sketch.*

   (i) **Work on a support-constrained face.** On the $\lambda$-on window $W = [t_1, t_2]$, Assumption B5 makes the penalized and the support-constrained problems equivalent; analyze the dynamics on that face while $t \in W$.

   (ii) **Uniformly correct gap signs.** By Lemma  B.2 (using B1) and Lemma  B.3 (using B1–B5), with probability at least $1 - \alpha/2$ all empirical gaps keep the correct sign throughout $W$.

   (iii) **Gate stabilization within $W$.** Conditioned on (ii) and using bounded gate temperature (B4), Lemma  B.4 with failure level $\alpha/2$ gives stabilization by some $t \in W$ provided $|W| \geq T_{\text{supp}}(\alpha/2)$. A union bound yields overall probability $\geq 1 - \alpha$.

   (iv) **Phase-B contraction.** For $t \geq t^*$ the usage term is constant on $\{S_t = S^*\}$; applying B6–B7 and Lemma  B.5 gives one-step contraction on the reduced objective.

**Roadmap.** Corollary  B.1 gives the high-level phase split: **Phase A** selects and stabilizes the support; **Phase B** optimizes on that fixed support with a linear-to-floor rate. The next section proves these sub-phase guarantees in order: Lemma  B.2 (margin concentration under B1), Lemma  B.3 (margin stability under the statewise quadratic), Lemma  B.4 (gate-stabilization horizon $T_{\text{sup}}(\alpha)$ under B1–B4 and the $\lambda$-on window of B5), and Lemma  B.5 (Phase-B contraction under B6–B7). We then combine them in §B.5 to obtain Theorem  B.6.

### B.4 SUB-PHASE CONVERGENCE

**Lemma B.2 (Uniform concentration of energy gaps over a finite window; uses B1–B3)).** *Let $W \subset \mathbb{N}$ be any finite index set (e.g., a $\lambda$-on window) and let $\{\theta_t : t \in W\} \subset \Theta$. For each component $j$ define*

$$g_j(X_o; \theta) := e_{(j)}^{\text{rej}}(X_o; \theta) - e_{(j)}^{\text{sel}}(X_o; \theta), \qquad \hat{\Delta}_j(\theta) := \frac{1}{M} \sum_{m=1}^{M} g_j(X_o^{(m)}; \theta), \qquad \Delta_j(\theta) := \mathbb{E}[g_j(X_o; \theta)].$$

*Assume B1–B3, and (as part of B3's regularity) that the centered variable $g_j(X_o; \theta) - \Delta_j(\theta)$ is sub-Gaussian with proxy $\kappa$ uniformly over $\theta \in \{\theta_t : t \in W\}$. Then, for every $\alpha \in (0, 1)$ there exists a universal constant $C > 0$ such that*

$$\Pr\left(\sup_{(t,j) \in W \times [N]} \left|\hat{\Delta}_j(\theta_t) - \Delta_j(\theta_t)\right| \leq C\kappa\sqrt{\frac{\log\left(\frac{2N|W|}{\alpha}\right)}{M}}\right) \geq 1 - \alpha.$$

*In particular, if B2 holds with margin $\gamma_0 > 0$ and*

$$\gamma_0 > 2C\kappa\sqrt{\frac{\log\left(\frac{2N|W|}{\alpha}\right)}{M}},$$

*then with probability at least $1 - \alpha$ the empirical gaps $\hat{\Delta}_j(\theta_t)$ have the same sign as the population gaps $\Delta_j(\theta_t)$ simultaneously for all $j \in [N]$ and all $t \in W$.*

*Proof.* For each fixed $(t, j)$, the sample mean $\hat{\Delta}_j(\theta_t)$ deviates from $\Delta_j(\theta_t)$ with sub-Gaussian tails $\leq 2\exp(-cMu^2/\kappa^2)$. Choose $u$ so that this tail probability equals $\alpha/(N|W|)$ and apply a union bound over $(t, j) \in W \times [N]$ to obtain the displayed bound (with $C = \sqrt{2/c}$). The sign conclusion follows because $|\hat{\Delta}_j - \Delta_j| \leq \gamma_0/2$ forces $\text{sign}(\hat{\Delta}_j) = \text{sign}(\Delta_j)$. The concentration inequality follows from standard sub-Gaussian tail bounds ( Vershynin (2018), Lemma 3.4.2). □

*Remark.* The margin scales as $O\left(\sqrt{\log(N|W|)/M}\right)$; large $M$ uniformly sharpens the gap estimates over all components and times in $W$, which is exactly what Phase-A stabilization (Lemma B.4) relies on.

**Lemma B.3** (**Robustness to the statewise quadratic, uses B2–B3**). *Let $W \subset \mathbb{N}$ be a finite window with parameters $\{\theta_t : t \in W\}$ and step sizes $\{\eta_t : t \in W\}$.*

*Decompose the training objective as*

$$f(\theta) = f_{\text{wo-se}}(\theta) + \lambda_{\text{se}} R_{\text{se}}(\theta), \qquad R_{\text{se}}(\theta) = \mathbb{E}\sum_{j=1}^{N}\left(e_{\text{sel}}^{(j)}(X; \theta)^2 + e_{\text{rej}}^{(j)}(X; \theta)^2\right),$$

*where $f_{\text{wo-se}}$ collects all terms except the statewise quadratic.*

*The update is*

$$\theta_{t+1} = \theta_t - \eta_t \nabla f(\theta_t) = \theta_t - \eta_t\left(\nabla f_{\text{wo-se}}(\theta_t) + \lambda_{\text{se}}\nabla R_{\text{se}}(\theta_t)\right), \qquad t \in W.$$

*Define the signed population margin*

$$m_j(\theta) := \begin{cases} \Delta_j(\theta), & j \in S^*, \\ -\Delta_j(\theta), & j \notin S^*, \end{cases} \qquad \left(m_j(\theta) \geq 0 \text{ encodes the correct sign}\right),$$

*with*

$$\Delta_j(\theta) = \mathbb{E}\left[e_j^{\text{rej}}(X_o; \theta) - e_j^{\text{sel}}(X_o; \theta)\right].$$

*Assume B2 (margin $\gamma_0 \geq 0$) and B3 (gap regularity), and suppose along $W$ we have local bounds*

$$\left|e_s^{(j)}(X; \theta_t)\right| \leq U, \qquad \left\|\nabla_\theta e_s^{(j)}(X; \theta_t)\right\| \leq G, \qquad s \in \{\text{sel}, \text{rej}\}, \; j \in [N], \; t \in W.$$

*Let $\bar{\eta} := \max_{t \in W} \eta_t$ and $B_{\text{se}} = 2NUG$. Then the incremental drift in margins due to the statewise quadratic over $W$ is bounded by*

$$\left|\hat{\Delta}_j(\theta_{t_1}) - \Delta_j(\theta_{t_2})\right|_{(se\text{-}term\ only)} \leq L_\Delta \lambda_{\text{se}} B_{\text{se}} \sum_{t=t_1}^{t_2-1} \eta_t \leq L_\Delta \lambda_{\text{se}} B_{\text{se}} |W| \bar{\eta}, \qquad (\star)$$

*for any $t_1 < t_2$ in $W$.*

*If, in addition, the population margins maintain a uniform buffer over the window,*

$$\inf_{(t,j)\in W\times[N]} m_j(\theta_t) \ \geq \ \tfrac{\gamma_0}{2}, \qquad \lambda_{\mathrm{se}} \ \leq \ \frac{\gamma_0}{2L_\Delta B_{\mathrm{se}}|W|\bar\eta},$$

*then $m_j(\theta_t) \geq 0$ for all $j \in [N]$ and all $t \in W$; i.e., the statewise quadratic cannot flip the correct sign of the margins anywhere in $W$.*

*Combining this with Lemma B.2 transfers the sign to the empirical gaps uniformly over $W$.*

*Proof sketch.* By the mean-value theorem and the Lipschitz constant from B3,

$$\big|\Delta_j(\theta_{t+1}) - \Delta_j(\theta_t)\big| \ \leq \ L_\Delta \, \|\theta_{t+1} - \theta_t\|.$$

The se-term contributes $\eta_t \lambda_{\mathrm{se}} \|\nabla R_{\mathrm{se}}(\theta)\|$ to each step, and

$$\|\nabla R_{\mathrm{se}}(\theta)\| \ \leq \ 2\sum_{j,s} \mathbb{E}\Big[\big|e_s^{(j)}\big| \cdot \big\|\nabla_\theta e_s^{(j)}\big\|\Big] \ \leq \ B_{\mathrm{se}}$$

by the $U, G$ bounds.

Summing over $t \in W$ and using $\sum_{t\in W} \eta_t \leq |W|\bar\eta$ yields $(\star)$. The buffer $\gamma_0/2$ therefore persists under the stated.

*Remark.* The lemma is deterministic: it isolates the se-term's effect and shows that, with $\lambda_{\mathrm{se}}$ small enough over $W$, that term cannot erode more than half of a pre-existing margin buffer. The buffer can be ensured by B2 together with staying in a small neighborhood (from B3 and standard small-step control); Lemma B.2 then supplies the empirical counterpart uniformly over $W$.

**Lemma B.4** (**Gating stabilization horizon in Phase A; uses B1–B4**). *Let $W = [t_1, t_2] \subset \mathbb{N}$ be a $\lambda$-on window (Assumption B5) with $\theta_t \in \Theta$ and step sizes $\{\eta_t : t \in W\}$. Assume B1–B3 and the weight choice in Lemma B.3 (so population margins keep the correct sign along $W$). During Phase A we evaluate the gate deterministically via the two-logit map $y = \sigma(\Delta/\tau_t)$ (i.e., without Gumbel resampling); the selected support at time $t$ is $S_t = \{\, j : \Delta_j(\theta_t) > 0 \,\}$. For any integer $L \geq 1$ define*

$$r_M(L) := C\kappa \sqrt{\frac{\log\!\big(\frac{2N|W|}{\alpha}\big)}{M}} \qquad and \qquad D(L) := L_\Delta \, \lambda_{\mathrm{se}} \, B_{\mathrm{se}} \, |W| \, \bar\eta,$$

*where $C, \kappa$ are the sub-Gaussian constants from Lemma B.2, $L_\Delta$ is the gap–Lipschitz constant (B3), $B_{\mathrm{se}} = 2NUG$ and $\bar\eta = \max_{t\in W} \eta_t$ are as in Lemma B.3. If*

$$r_M(L) + D(L) \ \leq \ \frac{\gamma_0}{2}, \tag{B.4.1}$$

*then, with probability $1 - \alpha$, any length-$L$ sub-window $W' \subseteq W$ enjoys*

$$\mathrm{sign}\big(\hat\Delta_j(\theta_t)\big) = \mathrm{sign}\big(\Delta_j(\theta_t)\big) \quad \text{for all } (t,j) \in W' \times [N],$$

*hence the deterministic gate selects the* same *support on all of $W'$:*

$$S_t \equiv S^* \qquad \forall t \in W'.$$

*In particular, if $W$ contains a contiguous block of length*

$$T_{\mathrm{supp}}(\alpha) := \min\left\{ \frac{\gamma_0}{4L_\Delta \lambda_{\mathrm{se}} B_{\mathrm{se}}|W|\bar\eta}, \ \ \frac{\alpha}{2N} \exp\!\Big(\frac{\gamma_0^2 M}{16C^2\kappa^2}\Big) \right\}, \tag{B.4.2}$$

*then (A) holds with a "half-buffer" split $r_M(T_{\mathrm{supp}}) \leq \gamma_0/4$ and $D(T_{\mathrm{sup}}) \leq \gamma_0/4$, so there exists $t^* \in W$ such that $S_t = S^*$ for all $t \in \big[t^*, \, t^* + T_{\mathrm{sup}}(\alpha) - 1\big]$.*

*Proof sketch.* Lemma B.2 gives the uniform-in-time statistical error $r_M(L)$ over any length-$L$ block; Lemma B.3 bounds the *deterministic drift* in margins caused by the statewise quadratic by $D(L)$. If (B.4.1) holds, the (population) margin buffer $\gamma_0$ cannot be eroded below zero on that block, and the empirical margins match their signs; with deterministic gating this fixes the support over the block. The explicit $T_{\mathrm{supp}}(\alpha)$ in (B.4.2) is a convenient sufficient choice obtained by splitting the buffer evenly.

*Remark.* Condition (B) shows two levers for shorter stabilization blocks (fewer steps to a fixed support):

- **Statistical term** $r_M(L) = O\left(\sqrt{\frac{\log(NL/\alpha)}{M}}\right)$: larger $M$ (more data) shrinks $r_M$, so a shorter block suffices.
- **Geometric term** $D(L) = O(\lambda_{\text{se}}\bar{\eta}L)$: smaller $\lambda_{\text{se}}$ or step sizes, or a smaller window length $L$, reduces drift; this matches our preliminary test's result that large $\lambda_{\text{se}}$ cannot destabilize convergence.

Empirically, in ablation study, increasing $M$ (or decreasing $\lambda_{\text{se}}\bar{\eta}$) should show faster Phase-A stabilization (fewer iterations to a fixed support).

**Lemma B.5** (**PL descent under bounded perturbations in Phase B; uses B6–B7**). *Fix $t^*$ from Corollary B.1 so the support is constant for $t \geq t^*$. On this fixed face let $f$ be the reduced objective (gate fixed). Assume B6 (L-smooth + PL with parameter $\mu > 0$ in a neighborhood of a minimizer) and B7 (bounded stochastic perturbations). Let $g_t$ denote the stochastic gradient used at step $t$, satisfying*

$$\mathbb{E}[g_t \mid \mathcal{F}_t] = \nabla f(\theta_t), \qquad \mathbb{E}\big[\|g_t - \nabla f(\theta_t)\|^2 \mid \mathcal{F}_t\big] \leq \nu^2,$$

*where $\nu^2$ collects mini-batch noise and the energy-space perturbation from Eqs. (6)–(7).*

*Then for any step size $\eta_t \in (0, 1/L]$,*

$$\mathbb{E}[f(\theta_{t+1}) - f^* \mid \mathcal{F}_t] \leq (1 - \mu\eta_t)(f(\theta_t) - f^*) + \tfrac{L}{2}\eta_t^2\nu^2. \tag{B.5.1}$$

*Consequences.*

- *With a constant step size $\eta \in (0, 1/L]$,*

$$\mathbb{E}[f(\theta_{t+1}) - f^* \mid \mathcal{F}_t] \leq (1 - \mu\eta)^{t-t^*}(f(\theta_{t^*}) - f^*) + \tfrac{L\nu^2}{2\mu}, \tag{B.5.2}$$

  *i.e., linear convergence to a variance floor.*

- *With a decreasing step size $\eta_t \downarrow 0$ and $\sum_t \eta_t = \infty$, the variance term vanishes and $\mathbb{E}[f(\theta_t)] \to f^*$.*

*Proof sketch.* $L$-smoothness gives $f(\theta_{t+1}) \leq f(\theta_t) - \eta_t\langle\nabla f(\theta_t), g_t\rangle + \tfrac{L}{2}\eta_t^2\|g_t\|^2$. Taking conditional expectation and using unbiasedness plus $\mathbb{E}\|g_t\|^2 \leq \|\nabla f(\theta_t)\|^2 + \nu^2$ yields

$$\mathbb{E}[f(\theta_{t+1}) \mid \mathcal{F}_t] \leq f(\theta_t) - \eta_t\left(1 - \tfrac{L\eta_t}{2}\right)\|\nabla f(\theta_t)\|^2 + \tfrac{L}{2}\eta_t^2\nu^2.$$

With $\eta_t \leq 1/L$ and PL, $\tfrac{1}{2}\|\nabla f\|^2 \geq \mu(f - f^*)$, which gives (B.5.1); (B.5.2) follows by iteration.

*Remark.* In **Phase A** the usage term sets the support; in **Phase B** that term is constant, so Lemma B.5 applies to **the same full loss** $f$ that is optimized in the implementation. The EB-select refinement noise from Eqs. (6)–(7) only enlarges $\nu^2$ (hence the variance floor $\tfrac{L\nu^2}{2\mu}$ in (B.5.2)); it does not change the limit point.

## B.5  MAIN RESULT

**Theorem B.6** (**Overall convergence under B1–B7**). *Assume B1–B7. Let $[t_1, t_2]$ be a window with $\lambda_t \geq \lambda^*$ and $\tau_t \leq \bar{\tau}$ for all $t \in [t_1, t_2]$. Suppose the window length $|[t_1, t_2]| \geq T_{\mathrm{supp}}(\alpha)$, where $T_{\mathrm{supp}}(\alpha)$ is the stabilization horizon of Lemma B.4. Then, with probability $1 - \alpha$, there exists $t^* \in [t_1, t_2]$ such that the support stabilizes,*

$$S_t = S^* \quad \textit{for all } t \geq t^*. \tag{B.6.1}$$

*For all $t \geq t^*$, the usage term is constant on the fixed-support face $\{S_t = S^*\}$. Let $f$ denote the reduced objective (gate fixed). If $f$ is L-smooth and satisfies the PL inequality with parameter $\mu > 0$ on a neighborhood of a minimizer (B6), and the stochastic update noise has bounded second moment $\nu^2$ (B7), then for any step size $\eta_t \in (0, 1/L]$,*

$$\mathbb{E}[f(\theta_{t+1}) - f^* \mid \mathcal{F}_t] \leq (1 - \mu\eta_t)(f(\theta_t) - f^*) + \tfrac{L}{2}\eta_t^2\nu^2, \qquad t \geq t^*. \tag{B.6.2}$$

*Unrolling gives, for all $t \geq t^*$,*

$$\mathbb{E}[f(\theta_t) - f^*] \leq (1 - \mu\eta)^{t-t^*}(f(\theta_{t^*}) - f^*) + \tfrac{L\eta\nu^2}{2\mu}. \tag{B.6.3}$$

*With a decreasing step size $\eta_t \downarrow 0$ and $\sum_t \eta_t = \infty$, and $\sum_t \eta_t^2 < \infty$, the variance term vanishes and $\mathbb{E}[f(\theta_t)] \to f^*$.*

*Proof sketch.*

(i) Lemma B.2 (B1–B3) gives a uniform $O\left(\sqrt{\frac{\log(N|W|/\alpha)}{M}}\right)$ bound on empirical–population gap error across the window.

(ii) Lemma B.3 ensures the statewise quadratic (weight $\lambda_{\mathrm{se}}$) cannot erode the margin buffer along the window.

(iii) Lemma B.4 then yields a stabilization horizon $T_{\mathrm{supp}}(\alpha)$; if $|[t_1, t_2]| \geq T_{\mathrm{supp}}(\alpha)$, we obtain (B.6.1).

(iv) After $t^*$, apply Lemma B.5 (B6–B7) on the fixed-support face to get (B.6.2)–(B.6.3).

# C  DETAILS OF CONSTRUCTING SYNTHETIC MCR DATASET

In synthetic datasets, components (base patterns) live in $\mathbb{R}^{512}$, mirroring the resolution of common chemical instruments. A pool of $N$ non-negative basis spectra are produced by sampling an orthonormal matrix and applying an element-wise absolute value, yielding mutually distinct yet chemically plausible components. For every mixture, the active component number $k$ is drawn uniformly from $\{K_{\min}, \ldots, K_{\text{high}}\}$ with $K_{\min} = 1$ and $K_{\max} = 4$, a range that reflects the limited number of dominant constituents typically encountered in laboratory samples. Concentrations are sampled uniformly from $[1, 10)$, ensuring that individual component signals remain on a comparable numerical scale and preventing any single component from dominating the mixture. The noiseless signal arises from summing these weighted components, after which zero-mean Gaussian noise is injected to reach a target SNR, thereby emulating instrument noise. This procedure generates spectra consistent with chemical intuition while preserving ground-truth components for quantitative evaluation of MCR methods. All experiments are executed on Ubuntu 24.04 LTS with two NVIDIA RTX A6000 GPUs (NVIDIA, Santa Clara, CA, USA).

---

**Algorithm 2** Mixed Data Sampling Process to Construct a Synthesized Dataset

---

**Input:** Candidate component matrix $S \in \mathbb{R}^{N \times d}$, number of mixed samples $M$, sampling range of component number in each mixture $K \in [K_{\min}, K_{\text{high}}]$, concentration range for each selected component $c \in [c_{\text{low}}, c_{\text{high}}]$

**Output:** Mixed synthesized data $X^d$

1: **for** $i = 1$ to $M$ **do**
2:  Sample the number of components: $k \sim \mathcal{U}\{K_{\min}, K_{\text{high}}\}$
3:  Randomly select a set of $k$ component indices $J_i \subset \{1, 2, \ldots, N\}$
4:  **for** each $j \in J_i$ **do**
5:   Sample concentration $c_j \sim \mathcal{U}(c_{\text{low}}, c_{\text{high}})$
6:  **end for**
7:  Compute the mixed signal:

$$X_i = \sum_{j \in J_i} c_j \cdot S_{j,:}, \quad \text{where } S_{j,:} \text{ is the } j\text{-th component}$$

8:  **if** SNR (dB) is specified **then**
9:   Compute $\sigma^2$ from $\|X_i\|_2^2$ and SNR (dB)
10:   Add $\varepsilon_i \sim \mathcal{N}(0, \sigma^2 I)$ to $X_i$
11:  **end if**
12: **end for**
13: **return** the matrix $X = [X_1, X_2, \ldots, X_M] \in \mathbb{R}^{M \times d}$

---

## D  BENCHMARK SEARCH WRAPPER FOR BASELINE MCR SOLVERS

To place each traditional MCR algorithm on equal footing with EB-gMCR, we wrap every solver in a simple component number search helper (Algorithm 3). Because the ground-truth component number ($C^*$) is known for the synthetic datasets, the wrapper probes a small band around that value and returns the best reconstruction it achieves. If the true component number were unknown —as in a real experiment—the baseline would need to extend the probe range and its runtime would grow unpredictably; for clarity we restrict the search to the nine ratios listed in $R_{search}$.

---

**Algorithm 3** Component number searching proxy for baseline MCR methods

---

**Require:** Observed data $D_o$, MCR solver initialized with $C$ components $f(\cdot, C)$, target reconstruction $R^2$ $\hat{R}$, True component number $C^*$, Searching ratios $R_{\text{search}} = \{0.80, 0.85, 0.90, 0.95, 1.00, 1.05, 1.10, 1.15, 1.20\}$
**Ensure:** Success $\in \{0, 1\}$, $R^2_{\text{best}}$, $C_{\text{sel}}$
1: Initialize: $R^2_{\text{best}} \leftarrow -\infty$, $C_{\text{sel}} \leftarrow$ None
2: **for** $s$ in $S$ **do**
3:    $C_{\text{try}} \leftarrow \lfloor s \cdot C^* \rceil$
4:    $R^2 \leftarrow f(D_o, C_{\text{try}})$                          ▷ run solver and compute $R^2$
5:    **if** $R^2 \geq R^2_{\text{best}}$ **then**
6:       $R^2_{\text{best}} \leftarrow R^2$
7:       $C_{\text{sel}} \leftarrow C_{\text{try}}$
8:    **end if**
9: **end for**
10: $success \leftarrow 1$ **if** $R^2_{\text{best}} \geq \hat{R}$ **else** 0
11: **return** $success, R^2_{\text{best}}, C_{\text{sel}}$

---

## E  SYNTHETIC-DATA EVALUATION DETAILS

Table 4 lists the exact mean ± 1 SD component numbers that form every point and ribbon in Figure 2. The rows cover all four dataset settings ($4N$ and $8N$ samples, each at 20dB and 30dB noise) and group the results by the $R^2$ checkpoint bands used for model selection. Each entry reflects the average over five independent draws, identical to those in the figure. Readers who wish to reproduce the curves, perform secondary analyses, or benchmark alternative solvers can extract the numeric values directly from this table. Table 3 lists the component number estimation and mixed signal reconstruction performance of EB-gMCR and benchmark methods (NMF, sparse-NMF, Bayes-NMF, ICA, and MCR-ALS). The success rate (succ.) defined as the fraction of runs that reached an $R^2$ no lower than EB-gMCR's own threshold (0.98 at 20dB, 0.99 at 30dB).

Table 3: Comparison of EB-gMCR with five baseline solvers on 4N synthetic mixtures.

| Components | Method | 20dB (success $R^2 \geq 0.98$) | | | 30dB (success $R^2 \geq 0.99$) | | |
|---|---|---|---|---|---|---|---|
| | | Succ. | $R^2$ | EC | Succ. | $R^2$ | EC |
| 16 | NMF | 0.33 | 0.978±0.002 | 18.0±1.1 | 1.00 | 0.995±0.002 | 17.0±1.4 |
| | Sparse-NMF | 0.00 | 0.761±0.034 | 13.7±2.0 | 0.00 | 0.776±0.031 | 13.4±1.9 |
| | Bayes-NMF | 0.18 | 0.978±0.002 | 18.3±0.9 | 1.00 | **0.996±0.002** | **17.6±1.4** |
| | ICA | 0.00 | 0.106±0.172 | 14.7±1.9 | 0.00 | 0.079±0.199 | 13.8±1.4 |
| | MCR-ALS | 0.80 | 0.981±0.003 | 16.4±0.9 | 0.92 | 0.994±0.008 | 16.1±1.4 |
| | EB-gMCR | - | **0.983±0.001** | **16.2±1.1** | - | 0.993±0.002 | 15.4±0.5 |
| 32 | NMF | 0.01 | 0.976±0.002 | 36.4±1.6 | 1.00 | 0.992±0.001 | 32.7±1.2 |
| | Sparse-NMF | 0.00 | 0.580±0.034 | 25.5±1.8 | 0.00 | 0.596±0.034 | 25.4±1.5 |
| | Bayes-NMF | 0.08 | 0.978±0.002 | 37.3±1.3 | 1.00 | **0.993±0.002** | **33.1±2.1** |
| | ICA | 0.00 | 0.230±0.123 | 30.4±3.5 | 0.00 | 0.131±0.188 | 29.1±2.9 |
| | MCR-ALS | 0.12 | 0.956±0.048 | 29.5±2.8 | 0.52 | 0.986±0.015 | 30.3±2.1 |
| | EB-gMCR | - | **0.982±0.001** | **31.4±0.9** | - | 0.992±0.001 | 31.4±2.2 |
| 48 | NMF | 0.00 | 0.972±0.003 | 55.8±1.6 | 0.83 | 0.991±0.001 | 53.9±2.7 |
| | Sparse-NMF | 0.00 | 0.524±0.019 | 38.8±2.2 | 0.00 | 0.536±0.019 | 38.9±3.0 |
| | Bayes-NMF | 0.03 | 0.977±0.002 | 56.2±1.4 | 1.00 | **0.992±0.001** | **49.5±1.8** |
| | ICA | 0.00 | 0.200±0.141 | 47.8±5.9 | 0.00 | 0.126±0.180 | 43.8±3.6 |
| | MCR-ALS | 0.00 | 0.870±0.139 | 41.5±3.7 | 0.11 | 0.923±0.108 | 42.1±3.2 |
| | EB-gMCR | - | **0.981±0.001** | **48.0±1.2** | - | 0.992±0.001 | 50.0±4.2 |
| 64 | NMF | 0.00 | 0.967±0.003 | 74.7±2.0 | 0.09 | 0.987±0.002 | 74.4±2.3 |
| | Sparse-NMF | 0.00 | 0.509±0.019 | 51.0±0.0 | 0.00 | 0.518±0.018 | 51.1±0.6 |
| | Bayes-NMF | 0.00 | 0.976±0.002 | 74.9±1.8 | 1.00 | **0.992±0.001** | **67.9±2.3** |
| | ICA | 0.00 | 0.215±0.114 | 63.3±8.3 | 0.00 | 0.172±0.143 | 58.5±4.7 |
| | MCR-ALS | 0.00 | 0.770±0.226 | 54.2±4.1 | 0.00 | 0.876±0.183 | 56.5±4.9 |
| | EB-gMCR | - | **0.981±0.001** | **64.8±2.6** | - | 0.991±0.001 | 74.4±5.0 |
| 96 | NMF | 0.00 | 0.955±0.004 | 113.3±2.9 | 0.00 | 0.977±0.003 | 113.7±2.4 |
| | Sparse-NMF | 0.00 | 0.504±0.015 | 76.3±1.6 | 0.00 | 0.515±0.015 | 76.1±1.0 |
| | Bayes-NMF | 0.00 | 0.974±0.002 | 114.2±1.9 | 1.00 | 0.991±0.001 | 108.0±3.8 |
| | ICA | 0.00 | 0.245±0.077 | 96.0±12.0 | 0.00 | 0.194±0.100 | 89.5±7.8 |
| | MCR-ALS | 0.00 | 0.782±0.237 | 86.4±6.9 | 0.00 | 0.882±0.041 | 85.9±6.2 |
| | EB-gMCR | - | **0.981±0.001** | **103.8±12.0** | - | **0.991±0.000** | **102.4±4.8** |
| 128 | NMF | 0.00 | 0.938±0.005 | 152.0±2.2 | 0.00 | 0.962±0.006 | 151.4±3.0 |
| | Sparse-NMF | 0.00 | 0.501±0.011 | 102.0±0.0 | 0.00 | 0.513±0.014 | 102.0±0.0 |
| | Bayes-NMF | 0.00 | 0.970±0.002 | 152.0±2.2 | 0.66 | 0.990±0.001 | 150.7±3.5 |
| | ICA | 0.00 | 0.275±0.065 | 129.3±13.7 | 0.00 | 0.235±0.085 | 120.0±9.9 |
| | MCR-ALS | 0.00 | 0.703±0.020 | 116.6±8.8 | 0.00 | 0.763±0.035 | 118.3±7.4 |
| | EB-gMCR | - | **0.981±0.000** | **147.8±14.9** | - | **0.991±0.000** | **137.6±3.8** |
| 160 | NMF | 0.00 | 0.923±0.005 | 191.2±2.7 | 0.00 | 0.945±0.005 | 190.8±2.9 |
| | Sparse-NMF | 0.00 | 0.500±0.012 | 128.0±0.0 | 0.00 | 0.513±0.011 | 128.0±0.0 |
| | Bayes-NMF | 0.00 | 0.967±0.002 | 191.4±2.2 | 0.18 | 0.988±0.002 | 191.3±2.3 |
| | ICA | 0.00 | 0.309±0.050 | 151.4±17.0 | 0.00 | 0.270±0.068 | 146.3±11.6 |
| | MCR-ALS | 0.00 | 0.445±0.055 | 140.0±10.9 | 0.00 | 0.480±0.080 | 142.6±10.9 |
| | EB-gMCR | - | **0.979±0.002** | **187.2±15.1** | - | **0.991±0.001** | **169.6±5.3** |
| 192 | NMF | 0.00 | 0.906±0.005 | 229.2±2.7 | 0.00 | 0.928±0.005 | 228.2±4.1 |
| | Sparse-NMF | 0.00 | 0.497±0.010 | 153.0±0.0 | 0.00 | 0.510±0.009 | 153.0±0.0 |
| | Bayes-NMF | 0.00 | 0.962±0.002 | 228.9±3.4 | 0.00 | 0.985±0.002 | 229.4±2.4 |
| | ICA | 0.00 | 0.316±0.048 | 171.4±17.2 | 0.00 | 0.285±0.054 | 170.7±12.7 |
| | MCR-ALS | 0.00 | -0.029±0.117 | 161.3±9.8 | 0.00 | -0.002±0.126 | 164.1±12.2 |
| | EB-gMCR | - | **0.979±0.000** | **213.5±11.6** | - | **0.989±0.002** | **191.8±4.6** |
| 224 | NMF | 0.00 | 0.893±0.004 | 266.9±3.3 | 0.00 | 0.914±0.004 | 267.2±2.8 |
| | Sparse-NMF | 0.00 | 0.497±0.009 | 268.0±0.0 | 0.00 | 0.510±0.009 | 268.0±0.0 |
| | Bayes-NMF | 0.00 | 0.957±0.002 | 267.7±1.9 | 0.00 | 0.981±0.002 | 267.2±2.8 |
| | ICA | 0.00 | 0.321±0.042 | 196.0±14.0 | 0.00 | 0.292±0.049 | 191.6±11.6 |
| | MCR-ALS | 0.00 | -0.676±0.164 | 185.9±9.2 | 0.00 | -0.617±0.150 | 186.4±10.2 |
| | EB-gMCR | - | **0.979±0.002** | **251.5±13.0** | - | **0.989±0.002** | **224.8±6.3** |
| 256 | NMF | 0.00 | 0.884±0.004 | 304.9±4.8 | 0.00 | 0.902±0.004 | 305.1±4.6 |
| | Sparse-NMF | 0.00 | 0.497±0.009 | 307.0±0.0 | 0.00 | 0.509±0.009 | 307.0±0.0 |
| | Bayes-NMF | 0.00 | 0.951±0.003 | 306.0±3.5 | 0.00 | 0.977±0.002 | 306.6±2.2 |
| | ICA | 0.00 | 0.341±0.031 | 212.8±9.0 | 0.00 | 0.307±0.036 | 214.3±9.2 |
| | MCR-ALS | 0.00 | -1.473±0.242 | 212.2±11.4 | 0.00 | -1.420±0.250 | 213.6±12.7 |
| | EB-gMCR | - | **0.978±0.003** | **286.7±6.4** | - | **0.990±0.000** | **248.8±1.1** |

Table 4: Exact numerical values underlying Fig. 2: mean $\pm$ 1 SD estimated component numbers returned by EB-gMCR for each dataset size ($4N$, $8N$) and noise level (20 dB, 30 dB).

| $R^2$ band | 16 | 32 | 48 | 64 | 96 | 128 | 160 | 192 | 224 | 256 |
|---|---|---|---|---|---|---|---|---|---|---|
| **Data size = 4N (SNR = 20dB)** | | | | | | | | | | |
| 0.97–0.975 | 15.2±1.6 | 29.8±0.4 | 44.4±0.5 | 58.6±1.1 | 88.8±3.0 | 122.6±6.5 | 156.2±4.3 | 184.0±3.7 | 225.6±6.3 | 260.4±6.6 |
| 0.975–0.98 | 15.8±1.3 | 30.4±0.5 | 46.0±0.7 | 61.4±1.3 | 92.8±4.9 | 131.4±6.9 | 166.0±6.8 | 192.4±3.0 | 234.0±6.0 | 265.4±5.6 |
| 0.98–0.985 | 16.2±1.1 | 31.4±0.9 | 48.0±1.2 | 64.8±2.6 | 103.8±12.0 | 147.8±14.9 | 187.2±15.1 | 213.5±11.6 | 251.5±13.0 | 286.7±6.4 |
| **Data size = 4N (SNR = 30dB)** | | | | | | | | | | |
| 0.97–0.975 | 13.6±0.5 | 27.0±0.7 | 39.8±0.4 | 52.4±1.1 | 79.6±2.1 | 110.2±1.9 | 138.6±4.3 | 170.0±2.0 | 200.4±3.8 | 234.4±4.4 |
| 0.975–0.98 | 14.2±0.4 | 27.6±0.5 | 40.4±0.5 | 53.8±0.8 | 82.0±2.4 | 113.2±0.8 | 141.2±3.0 | 173.2±1.8 | 202.8±2.5 | 237.2±2.6 |
| 0.98–0.985 | 14.4±0.5 | 28.2±0.8 | 41.8±0.4 | 55.2±1.1 | 85.2±2.8 | 116.0±1.9 | 145.2±2.2 | 176.2±0.8 | 207.0±2.0 | 239.2±1.9 |
| 0.985–0.99 | 14.8±0.4 | 29.2±0.8 | 43.2±0.4 | 57.2±1.1 | 88.4±1.9 | 120.8±1.1 | 149.2±1.6 | 180.0±1.0 | 211.2±1.5 | 242.8±1.3 |
| 0.99–0.995 | 15.4±0.5 | 31.4±2.2 | 50.0±4.2 | 74.4±5.0 | 102.4±4.8 | 137.6±3.8 | 169.6±5.3 | 191.8±4.6 | 224.8±6.3 | 248.8±1.1 |
| **Data size = 8N (SNR = 20dB)** | | | | | | | | | | |
| 0.97–0.975 | 20.4±1.9 | 40.0±3.7 | 53.8±2.9 | 65.0±2.2 | 93.4±0.9 | 128.2±2.5 | 166.0±4.8 | 200.2±5.5 | 237.8±10.4 | 272.4±6.3 |
| 0.975–0.98 | 20.2±1.6 | 39.8±4.4 | 53.8±2.9 | 65.6±1.9 | 96.4±0.9 | 142.6±11.0 | 187.0±10.0 | 225.0±9.0 | 260.8±9.9 | 301.0±6.4 |
| **Data size = 8N (SNR = 30dB)** | | | | | | | | | | |
| 0.97–0.975 | 14.4±0.5 | 31.2±0.8 | 44.4±1.7 | 57.0±0.7 | 85.8±0.8 | 114.8±1.8 | 147.2±1.9 | 178.4±1.7 | 210.4±0.7 | 241.6±2.3 |
| 0.975–0.98 | 14.8±0.4 | 31.2±0.8 | 44.8±1.3 | 58.2±0.6 | 89.2±2.5 | 117.4±1.5 | 150.0±1.9 | 181.6±1.1 | 213.2±3.3 | 245.0±1.1 |
| 0.98–0.985 | 15.0±0.4 | 31.2±0.8 | 45.4±0.9 | 59.2±0.7 | 90.6±2.2 | 120.6±1.1 | 153.0±1.9 | 185.0±1.4 | 216.4±2.3 | 248.0±1.1 |
| 0.985–0.99 | 15.4±0.5 | 31.2±0.5 | 46.0±0.7 | 61.0±0.5 | 92.2±1.6 | 124.0±1.6 | 156.0±1.2 | 187.2±1.1 | 220.0±1.3 | 251.6±0.5 |
| 0.99–0.995 | 16.0±0.5 | 32.4±0.5 | 47.4±0.5 | 66.0±0.5 | 96.2±0.8 | 128.0±1.6 | 159.6±1.1 | 191.8±0.4 | 223.0±0.8 | 255.0±1.4 |

Table 5: Real-data decomposability. $R^2$ can be negative under our definition; see §5.2.

| Solver | Carbs ($N = 3$, $M = 22$) | | | | NIR ($N = 2$, $M = 166$) | |
| | $R^2(D)\uparrow$ | RMSE($C$)$\downarrow$ | Cos.($S$)$\uparrow$ | RMSE($S$)$\downarrow$ | $R^2(D)\uparrow$ | RMSE($C$)$\downarrow$ |
|---|---|---|---|---|---|---|
| NMF | 0.963 | **0.270** | 0.596 | 5.957 | 0.876 | 32.457 |
| Sparse-NMF | 0.963 | **0.270** | 0.597 | 5.951 | -131.423 | 42.325 |
| Bayes-NMF | 0.962 | 0.272 | 0.593 | 5.963 | 0.888 | 34.062 |
| ICA | -6.206 | 0.437 | 0.401 | 6.752 | -66.987 | 23.467 |
| MCR-ALS | **0.964** | 0.287 | **0.618** | **5.809** | -25.167 | 27.630 |
| EB-gMCR | **0.964** | 0.281 | **0.614** | **5.844** | **0.910** | **24.493** |

## F  REAL-DATA EVALUATION DETAILS

paragraphDatasets. **Carbs**: $M = 22$, $\mathbb{R}^{1401}$, $N = 3$ non-negative sources; linear mixing with noise. **NIR**: $M = 166$, $\mathbb{R}^{235}$, signals include negatives; two-analyte concentrations are known while components are unknown.

**Baselines and preprocessing.** Baselines follow the adaptive rank sweep of Smith et al. (2019), increasing $N$ from 1 until $R^2$ stabilizes (cap 10). For methods constrained to nonnegativity, we apply a sign-splitting transform (separate $\pm$ parts, flip the negative part's sign, concatenate to a nonnegative $2d$ vector), then invert after fitting. EB-gMCR initializes 128 candidate components and stops when selection energy and the active set stabilize.

**Metrics and alignment.** We align components and concentrations via linear assignment. Reported metrics: coefficient of determination $R^2(D)$ ↑, Cos.($S$) ↑, RMSE($S$) ↓, RMSE($C$) ↓, nMSE($C$) ↓.

**Table 5** Per-method values are listed at the selected checkpoints; missing curves or negative $R^2$ indicate failure to achieve a reasonable fit. Scripts in the review bundle reproduce the table from saved outputs.

