# OpenReview forum: "EB-gMCR: Energy-Based Generative Modeling for Signal Unmixing and Multivariate Curve Resolution"
_ICLR.cc/2026/Conference — Submitted to ICLR 2026_

### Official Review · Reviewer_TaiB · 2025-10-29

**Soundness:** 3
**Presentation:** 2
**Contribution:** 2
**Rating:** 4
**Confidence:** 4

**Summary:**

The paper proposes EB-gMCR (Energy-Based Generative Modelling for Signal Unmixing and Multivariate Curve Resolution). This framework reformulates classical multivariate curve resolution (MCR) as a generative process rather than a matrix factorisation problem. The authors introduce an energy-based adaptive gating module (EB-select) that automatically selects a minimal subset of source components while maintaining reconstruction fidelity. This approach aims to overcome limitations of conventional MCR methods, such as the need to predefine the number of components, lack of scalability, and difficulty in incorporating domain constraints.
Experiments on synthetic benchmarks (up to 256 components) and two real spectral datasets demonstrate strong reconstruction accuracy and correct component-count recovery compared with NMF, sparse-NMF, ICA, and MCR-ALS baselines. The authors also provide detailed convergence proofs, ablations, and reproducibility statements.

**Strengths:**

* Recasts MCR as a generative problem, conceptually unifying signal unmixing and energy-based learning.

* Introduces a differentiable hard-selection mechanism (EB-select) enabling data-driven component pruning.

* Provides comprehensive theoretical derivations and convergence analysis.

* Demonstrates strong empirical results on spectral datasets and includes full reproducibility materials.

* A meaningful step forward for scalable unmixing and spectral decomposition within chemistry and materials analysis.

**Weaknesses:**

*  The claimed novelty is modest; the paper applies energy-based modelling (EBM) and sparse gating techniques, both well-studied concepts. For example, the recent review “Hitchhiker’s guide on Energy-Based Models” (Carbone, 2024) surveys EBMs and situates them among VAEs, flows and GANs, showing energy-based methods are mature and widely applied. Meanwhile, sparse gating or mixture-of-experts style selection mechanisms are well established (see e.g., Mixture-of-Experts architecture surveys).  As a result, the contribution here is primarily domain adaptation (spectral unmixing) rather than a fundamentally new generative modelling algorithm or gating mechanism.

* Although the paper claims plug-in adaptation across domains, validation is confined to chemical spectra. There are no results for real hyperspectral imaging datasets or other modalities.

* The method claims to handle thousands of components, yet experiments only reach 256 latent sources.

* Metrics focus on 𝑅2 and component count recovery; no generative metrics (likelihood, uncertainty, sample diversity) or comparisons to VAE/score-based baselines are reported.

* Despite claims of “hands-free” component discovery, λ and temperature schedules require manual tuning.

* Dense notation, minimal intuition for EB-select and gating dynamics.

**Questions:**

* How sensitive is EB-gMCR to the choice of the λ coefficients and temperature decay schedule? Can the authors provide quantitative ablation on these hyperparameters?

* Can the proposed framework be extended to nonlinear or multimodal unmixing tasks (e.g., image or audio mixtures)?

* How does EB-gMCR perform under correlated component spectra or severe noise (>30 dB)?

* Could the authors clarify how EB-select differs fundamentally from previous sparse or Concrete-gated EBMs beyond its application domain?

* Would evaluating on benchmark generative datasets (e.g., MNIST mixtures, speech sources) help demonstrate generality

---

> ### Author Response · Authors · 2025-11-20
> **Response to Reviewer TaiB.**
>
> We thank the reviewer for their time. However, we must respectfully note several misunderstandings about the problem scope. We recommend reviewing our official comment, which clarifies the MCR problem with a concrete example.
>
> **Core Clarification:** This work addresses **signal decomposition**, not generative synthesis. The appropriate comparisons are matrix factorization methods (NMF, ICA, MCR-ALS), not VAEs or diffusion models.
>
> **On the weaknesses:**
>
> **W1 (Novelty):** Our contribution is solving a fundamental MCR bottleneck: eliminating manual component number search. Traditional methods require exhaustive search over $N$ values (Table 1: 0.1–26 hours extrapolated for interval=1 at N=256). EB-gMCR discovers $N$ automatically in one run (2–3 hours) and produces a reusable model—MF methods must re-optimize for each new dataset.
>
> On sparse gating: MoE sparsity is an optimization choice for computational efficiency; modern sparse MoE requires auxiliary load balancing mechanisms. MCR sparsity reflects physical reality—a component either exists in a sample or doesn't (fixed at data collection). Soft gates violate this: you cannot have "0.3 of a chemical compound" in a mixture.
>
> **W2 (Validation scope):** We validated on chemical spectra because that is the MCR domain. "Hyperspectral imaging" typically uses different formulations (see hyperspectral unmixing literature).
>
> **W3 (Scale):** The paper explicitly states experiments reach **N=256** (abstract line 9, Section 5.1). We initialize with 1024 candidates that EB-gMCR prunes. The claim "only reach 256" is factually incorrect.
>
> **W4 (Metrics):** $R^2$ is the standard MCR metric (de Juan & Tauler, 2021). Generative metrics (likelihood, sample diversity) are inappropriate for decomposition—we don't synthesize mixtures; we infer components from observed ones. This is like asking why segmentation papers don't report FID scores.
>
> **W5 (Hyperparameters):** Same settings across all experiments (N=32–256, 20–30dB, 4N–8N, synthetic and real; Fig. 2, Tables 1–3). $\lambda$ is a Lagrange multiplier (auto-adjusted). Temperature uses fixed bounds. Only user-specified parameter is $R^2$ checkpoint band (quality target, not algorithmic hyperparameter). See Reviewer NBQX response for details.
>
> **W6 (Clarity):** We substantially revised mathematical presentation and added convergence analysis (Section 4.5, Appendix B).
>
> **On the questions:**
>
> **Q1:** Addressed in W5 and Reviewer zgrb response. Table 2 shows robustness.
>
> **Q2 (Nonlinear extension):** Framework applies to fixed-pattern unmixing (components retain signatures across mixtures). Audio with speaker-dependent patterns or general images don't satisfy this. We state this scope explicitly (Section 1, 5.4).
>
> **Q3 (Noise/correlation):** 20dB is more severe than 30dB by definition. We test both (Fig. 2). Component correlation doesn't affect MCR—components remain distinct if concentration profiles differ. HSIC prevents basis collapse, not correlation.
>
> **Q4 (Concrete-gated EBMs):** We're unfamiliar with this as a specific prior method. If the reviewer can provide a citation, we'll discuss the relationship. Gumbel-softmax (Jang et al., 2016; Maddison et al., 2016) is a gradient estimator, not an EBM architecture.
>
> **Q5 (Benchmark datasets):** MNIST mixtures and speech sources don't follow MCR structure (fixed-pattern linear superposition with unknown component count). We're not aware of a standard "MNIST mixtures" unmixing benchmark. If the reviewer can specify the dataset, we'll assess relevance.
>
> **Summary:** Several points stem from misunderstanding the problem class. MCR is decomposition/inverse problem, not synthesis. We compare against appropriate baselines (matrix factorization). The contributions—automatic component discovery and reusable decomposition—address real MCR workflow challenges.

---

> > ### Comment · Reviewer_TaiB · 2025-11-26
> >
> > Thank you for the detailed response. It is good to see the clarifications, and the additional explanations do help situate the intent and scope of the work. A few points may help align our perspectives.
> >
> > 1. Problem Framing
> >
> > My initial review did not intend to conflate decomposition with generative synthesis. The references to VAEs, score-based models, or generative metrics were meant to situate the method within the broader landscape of generative modelling terminology used in the paper. Since the paper introduces gMCR as a generative formulation and positions EB-gMCR as an energy-based generative solver, it was natural to ask where the approach sits relative to established EBM practice, and whether standard generative diagnostics could be informative. At the same time, the title, abstract, and Section 3 explicitly frame the method as “generative modelling” and a “data-generating formulation” of MCR, with EB-gMCR presented as an “energy-based generative solver” that learns a reusable generative model. My original comments about generative metrics and baselines were not intended to force a comparison to image-generation models, but to question where this work sits within the broader EBM/generative-modelling landscape that the paper itself invokes. Please soften the “generative modelling” framing (and position this more clearly as an inverse problem/decomposition method with an energy-based training objective).
> >
> > 2. Novelty and relation to existing EBM / sparse-gating work
> >
> > I appreciate the clarification that the core practical contribution is eliminating manual component-number search and enabling reusable decomposition within the MCR workflow. That is a meaningful domain contribution.
> >
> > From an ML/ICLR point of view, however, the methodological ingredients remain quite standard: energy-based parameterisation, Gumbel-softmax / Concrete relaxation for discrete gates, sparsity/usage penalties, and HSIC-style diversity regularisation. The rebuttal does not really change that assessment; instead, it clarifies the physical interpretation of sparsity in the MCR setting.
> >
> > I would encourage the authors to be more explicit in the paper about what is not new (in terms of EBM machinery and gating) and to clearly delimit the novelty to:
> >
> > * The particular way these ingredients are assembled for MCR, and
> >
> > * The contribution to large-scale MCR workflows (automatic component discovery + model reuse), rather than a fundamentally new EBM or gating architecture.
> >
> > This would also help align with Reviewer NBQX’s and zgrb’s comments on heuristic design and notation, and reduce the risk of overselling conceptual novelty.
> >
> > 3. Scalability and the “thousands of components” narrative
> >
> > The rebuttal states that my remark that the method “only reaches 256 components” is “factually incorrect” because the initial pool is 1024 candidates and the method prunes down to ≈256 effective components. I think this slightly misses the point.
> >
> > * The paper motivates use cases with “hundreds or thousands of candidate components” and positions EB-gMCR as a solver that scales to such regimes.
> >
> > * The experiments, however, demonstrate correctness and scaling up to true component numbers of at most 256 on synthetic data, and 2–3 components on real data.
> >
> > I have no objection to using synthetic data for scale (this is standard when ground truth is unavailable), but I still see a gap between the narrative “thousands of components” and the actual evaluated regime. If the authors want to keep the stronger scaling claims, I would suggest:
> >
> > * Either adding at least one synthetic setting with a true component count substantially beyond 256 (even without real data), or
> >
> > * Toning down some of the language around “thousands of components” and clearly framing 256 as the demonstrated regime, with larger scales left as future work.
> >
> > Relatedly, as Reviewer Ljcq and zgrb both noted, the real-data experiments with N=2–3 do not probe the claimed high-component advantage, so the paper should be careful not to present them as evidence of scaling.
> >
> > “Hands-free” discovery vs hyperparameter dependence
> >
> > 4. The response argues that λ is auto-adjusted, temperature uses fixed bounds, and the only user-specified parameter is the R² checkpoint band (as a “quality target, not an algorithmic hyperparameter”). In practice, though, the behaviour of the method depends on the interaction between:
> >
> > The λ schedule (which couples reconstruction loss and component usage),
> >
> > The temperature schedule (which controls how hard the gate becomes), and
> >
> > The choice of R² bands and early-stopping / checkpointing rules.
> >
> > The ablation table is helpful and does show some robustness, but it does not fully answer the original question about sensitivity: for example, how often does the method under- or over-estimate the component count if the R² band is varied, or if the temperature decay is slowed/accelerated? How often do different runs with the same settings converge to different EC values?

---

> ### Comment · Reviewer_TaiB · 2025-11-26
>
> --- Continue ---
>
>
> 5. Restricting validation
>
> Restricting validation to chemical spectra is reasonable given the stated domain. My comment regarding hyperspectral datasets was aimed at testing the claimed generality rather than suggesting a change in domain. Your explanation in the response makes these boundaries explicit.
>
> 6. EB-select vs previous Concrete-gated EBMs
>
> On the question about “Concrete-gated EBMs,” the rebuttal says the authors are unfamiliar with such a specific prior and emphasises that Gumbel-softmax is a gradient estimator, not an architecture. This is fine, but it does not really address the underlying concern: what, if anything, is architecturally or algorithmically new about EB-select relative to standard energy-parameterised gating with Concrete relaxation?
>
> If the answer is “we use a fairly standard Gumbel-softmax gate plus an energy regulariser in a specific MCR context,” that is perfectly acceptable, but then the paper should avoid language suggesting that EB-select is fundamentally new as a gating mechanism. Clarifying this would also help resolve the tension between the authors’ emphasis on physical interpretation and the reviewers’ emphasis on ML novelty.

---

> > ### Author Response · Authors · 2025-11-27
> > **Response to Reviewer TaiB.**
> >
> > Thank you for the thoughtful engagement. Your clarifications have helped us better position the work.
> >
> > **1. Problem framing:**
> > We agree and have revised the manuscript to explicitly frame this as solving an **inverse problem** on a generative process model. Specifically:
> > - The contribution now states: "We reformulate MCR as an inverse problem on a generative process (gMCR), and EB-gMCR learns a reusable solver that eliminates re-computation when analyzing new samples."
> > - Section 4 opening now reads: "Despite the natural forward formulation of gMCR, the reality of signal unmixing is that we must solve it as an inverse problem. EB-gMCR solves this inverse problem (inferring components from observed mixtures) using energy-based modeling."
> > This clarifies that gMCR describes the forward process (components → mixtures), while EB-gMCR solves the inverse decomposition (mixtures → components).
> >
> > **2. Novelty and relation to existing work:**
> >
> > We acknowledge that energy-based parameterization, Gumbel-softmax, sparsity penalties, and HSIC are established ML techniques (cited appropriately). The paper focuses on explaining **why each component is necessary for solving MCR** —why hard selection matters, why energy regularization ensures convergence, why component usage penalties enable automatic N discovery. Our goal is demonstrating that this design solves the MCR problem effectively, not claiming methodological novelty for its own sake.
> >
> > We've refined the description of EB-select to clarify it builds on Gumbel-softmax while adding energy space regularization for MCR-specific requirements. The emphasis throughout is on problem-solving utility rather than architectural novelty.
> >
> > **3. Scalability narrative:**
> >
> > We've revised all scale references:
> > - **"Thousands" language:** Now limited to describing application contexts (library searches with hundreds of candidates), not experimental claims. Changed "1000+ components" to "oversize candidate pools."
> > - **Experimental descriptions:** Now clearly separate synthetic (scales to N=256) from real data (validates correctness at N=2-3 with competitive reconstruction). The conclusion states: "Synthetic experiments validate scalability with accurate component recovery up to N=256, while real datasets (Carbs, NIR) validate correctness with proper identification and competitive reconstruction."
> >
> > We don't present N=2-3 real dataset results as scaling evidence.
> >
> > **4. Hyperparameter sensitivity:**
> >
> > Figure 2a-2d shows $R^2$ band sensitivity with error bars across 5 runs. Regarding your specific questions:
> > - **$R^2$ bands:** election is inherently domain-dependent—users consider expected noise in their data. We acknowledge that inappropriately high $R^2$ thresholds cause overestimation in high-noise scenarios. This is a practical consideration requiring domain knowledge.
> > - **Temperature:** The critical design is the train/eval gap (training $\tau ≥ 0.4$, evaluation  $\tau = 0.01$). We use temperature decay to 0.4 because higher initial temperatures facilitate faster exploration in early training, but preliminary tests show the method remains stable even with fixed  $\tau = 0.4$ throughout. Systematic schedule ablation is difficult without a clear optimization target.
> > - **EC variance:** Figure 2 error bars demonstrate stability across random initializations.
> > More extensive systematic ablation would strengthen the analysis, which we note as valuable future work.
> >
> > **6. EB-select vs. prior work:**
> >
> > We clarify that "concrete-gated EBM" doesn't appear as a method name in the literature. Regarding novelty, we acknowledge in our response to point #2 which components are established and which aspects are specific to our design.
> >
> > The MCR problem becomes challenging even at moderate scale—traditional methods show performance degradation starting around N=100, where reconstruction quality drops and search becomes unreliable. Our framework maintains stable performance to N=256 with results that align well with theoretical expectations (Fig. 2). This gap—from where traditional methods degrade to where EB-gMCR operates stably—represents a meaningful advance in numerical analysis of real-world signals.
> >
> > **What we believe is novel for MCR:**
> > - Explicit regularization of the energy (pre-softmax) space, not just using Gumbel-softmax as a black-box gradient estimator
> > - Convergence guarantees via energy dynamics analysis (Lemma B.3)
> > - Integration enabling automatic N discovery and model reusability
> >
> > The individual components are established—physics concepts are centuries old, ML techniques are standard. But the assembly addresses a real gap: it solves MCR at scales where previous methods fail, and does so in ways that match our intuition about how the problem should behave.
> >
> > We believe the framework can scale further in principle, though practical constraints limited our experiments to N=256. The contribution is demonstrating effective large-scale MCR through this particular combination of techniques.

---

### Official Review · Reviewer_NBQX · 2025-10-31

**Soundness:** 3
**Presentation:** 2
**Contribution:** 2
**Rating:** 2
**Confidence:** 3

**Summary:**

The paper proposes a generative model for signal unmixing that includes a generic aggregation function, concentration generator, and component patterns, all of which are implemented by deep learning networks. The model is trained via minimization of an energy function that accounts for correlation of the signal components via a kernel embedding, component usage, and selection energy.

**Strengths:**

The generic nature of the components of the mixing model makes the approach amenable to a variety of applications.

**Weaknesses:**

The approach uses a combination of heuristics (sparsity prior, component activations, $\ell_1$ norm penalty on component weights) but does not provide much motivation for the choices. Furthermore they are sourced from existing approaches.

There are multiple instances of notation not being defined (as detailed in "Questions").

A comparison of computation time with unmixing baselines that do not require training is missing.

**Questions:**

What is $\omega$ in eqs. (2-3)? It does not appear to be considered anywhere else.

In lines 201 and later, what is $X_o$/$X_0$?

In line 226, is $|{\mathbf E}\|_2^2$ supposed to be $\|{\mathbf E}(\omega)\|_2^2$?

What is $E_{f_e}$ in eq. (6)? Is it different than ${\mathbf E}(\omega)$?

What is the "coefficient of determination $R^2$"? Why is it used as a metric in Section 5? What is the "EC" metric?

---

> ### Author Response · Authors · 2025-11-20
> **Response to Reviewer NBQX**
>
> We sincerely appreciate the reviewer's feedback on mathematical clarity and result presentation. We have revised the manuscript to address these concerns. We also recommend reviewing our official comment for context on the MCR problem and the challenges we address.
>
> **Regarding the weaknesses:**
>
> **On heuristic design choices (W1):** We acknowledge that individual components (sparsity penalties, energy regularization, Lagrange multipliers) are not mathematically novel. Our contribution is **reformulating MCR as a generative process (gMCR)** and developing an energy-based solver that addresses two fundamental MCR bottlenecks:
>
> 1. **Eliminates manual component number search:** Traditional MCR methods require exhaustive search over candidate $N$ values, while EB-gMCR automatically discovers $N$ through energy-based selection in a single run.
>
> 2. **Enables reusable decomposition:** Traditional MF-based methods must re-optimize the entire factorization for each new dataset. EB-gMCR's generative formulation learns a reusable model that can rapidly decompose new observations without retraining—critical for high-throughput applications.
>
> The framework is heuristic but demonstrates strong empirical performance: accurate component recovery across N=16–256 (Fig. 2), competitive decomposability on real datasets (Fig. 3, Table 5), and successful scaling to regimes where baselines become unreliable (N>100).
>
> **On notation issues (W2):** We have systematically revised all notation. Specific corrections are detailed in our responses to Q1–Q4 below.
>
> **On computation time comparison (W3):** We have added Table 1, comparing wall-clock time between EB-gMCR and baselines at search intervals of 8, 16, and 32. While this comparison is asymmetric (baselines search upward from small $N$; EB-gMCR prunes from N=1024).
>
> **Regarding the questions:**
>
> **Q1 ($\omega$ in Eqs. 2-3):** Now precisely defined in the revised manuscript (around L131). It denotes a realization (sample) from the generative process—essentially indexing which mixture observation we're considering.
>
> **Q2 ($X_o$ vs $X_0$ in line 201+):** We apologize for this inconsistency. All instances have been corrected to $X_o$ throughout the manuscript, where "o" denotes "observed" data.
>
> **Q3 ($\|E\|_2^2$ vs $\|E(\omega)\|_2^2$ in line 226):** The original manuscript conflated two uses of "E"—measurement noise (Eq. 1) and selection energy. We have resolved this: measurement noise is now consistently denoted $\varepsilon$ (epsilon), while $E$ refers exclusively to the selection energy in the EB-select module. The notation in Eq. 13 is correct as written: $\|E\|_2^2$ refers to the selection energy regularization term.
>
> **Q4 ($E_{f_e}$ in Eq. 6 vs $E(\omega)$):** These refer to the same quantity—the selection energy output by the EB-select module $f_e$. We have unified the notation in the revised manuscript for consistency.
>
> **Q5 (Coefficient of determination $R^2$ and EC metric):** We have added an "Evaluation Metrics" section at the beginning of Section 5. In brief:
>
> - **$R^2$ (coefficient of determination):** The standard metric in MCR literature for assessing decomposition quality. Even under the linear mixing assumption, measurement noise and the unknown true forms of components and concentrations mean perfect reconstruction is impossible. $R^2$ measures how well the learned factorization reconstructs the observed data—the only ground truth available in real applications. Values near 1 indicate the decomposition effectively captures the mixing structure.
>
> - **EC (Estimated Component number):** Counts components with selection probability exceeding our threshold ($\tau_{\text{thres}} = 0.9999994$, corresponding to 5σ confidence). This metric evaluates EB-gMCR's ability to automatically discover the correct component count—a core contribution of our method. Baseline methods do not report EC because they require $N$ as a fixed input parameter.
>
> The EC metric directly measures what we believe is the central contribution: eliminating manual component number search through automatic energy-based discovery.

---

> > ### Comment · Reviewer_NBQX · 2025-11-21
> > **Thank you for your response**
> >
> > Thank you for adding information on the evaluation metrics and for highlighting the performance of the proposed approach as a strength. I plan to add that in a revised version of the review.
> >
> > I stil have some concerns on clarity of notation. There is likely a typo in line 243: |E||_2^2 with bolded E, which I cannot find elsewhere in the paper. I also wasn't able to find a formal definition (e.g., an equation) for E_{fe}. It is not clear if E_{fe} is a matrix or a finite energy function given the use of ||E_{fe}||_2^2.

---

> > > ### Author Response · Authors · 2025-11-22
> > > **Notation Corrections in Revised Manuscript**
> > >
> > > Thank you for your constructive feedback and for planning to revise your review.
> > >
> > > **Notation corrections:**
> > >
> > > Line 242 (and other bold notation in §4.2): Fixed all bolded notation inconsistencies.
> > >
> > > **E definition:** We have revised §4.2 to explicitly define $E \in \mathbb{R}^{N \times 2}$ as the selection energy matrix. Figure 1b shows the computation flow, and the regularization term $\|E\|_2^2$ is properly contextualized.
> > >
> > > We believe the revised EB-select paragraph improves clarity and addresses your mathematical presentation concerns.
> > >
> > > Thank you for your careful review and willingness to re-evaluate.

---

### Official Review · Reviewer_zgrb · 2025-10-31

**Soundness:** 2
**Presentation:** 2
**Contribution:** 3
**Rating:** 4
**Confidence:** 2

**Summary:**

This paper proposes EB-gMCR, an approach that reformulates the classical MCR problem as a data generative process (gMCR). The main innovation is an energy-based solver that automatically discovers the minimal component set needed for signal reconstruction without requiring a pre-specified number of components. The method starts with an oversized pool of candidate components and uses a differentiable "EB-select" gating network to retain only necessary components while estimating their concentrations. The authors validate their approach on synthetic benchmarks with up to 256 components and two real spectral datasets (carbohydrate Raman and NIR bioethanol).

**Strengths:**

* Overall, the writing is quite clear and logically structured (except for the mathematical formalism; see below).
* The reformulation of MCR as a generative process (gMCR) is innovative. Traditional MCR is typically framed as matrix factorization requiring a user-specified number of components, while this work provides a principled way to learn both the component set and their mixing patterns simultaneously.
* The ability to handle pools of 1000+ candidates is impressive and addresses real-world needs.
* The convergence analysis (Theorem B.6 and supporting lemmas) provides theoretical grounding for the two-phase learning dynamics, though the presentation could be clearer.
* The plug-in of domain constraints (non-negativity, nonlinear mixing) without requiring solver redesign is a  practical advantage over existing methods.

**Weaknesses:**

* Only two real datasets are tested, both relatively simple (N=3 and N=2 components). The method's performance on more complex real-world mixtures isn't shown.
* The method introduces several hyperparameters (λ weights, temperature τ, R² bands for checkpointing) whose selection process and sensitivity are not thoroughly discussed. Sensitivity or instability w.r.t. those parameters could be neck breaking for many more complex problems.
* The mathematical formalism is quite sloppy and therefore sometimes difficult to follow. E.g. what kind of elements are D, C, S and E? Assumptions and conclusions aren't strictly separated (see questions below).

**Questions:**

* All mathematical statements in the appendix contain proof sketches. Does that mean you have a checked proof for those statements, but you're not spelling out the proof? If so, why aren't you showing the full proof?
* Fig. 2 and 3: Are the R2 numbers on test or train data?
* Eq. 3: is this an assumption or a condition that is required? It seems like this does not follow from E. 2 in general.
* Eq. 5: Is the approximation an assumption or why would this be true? The expectation is in [0, 1] in general whereas the function maps to {0,1}.

---

> ### Author Response · Authors · 2025-11-20
> **Response to Reviewer  zgrb.**
>
> We sincerely appreciate the reviewer's feedback on mathematical presentation. We have carefully revised the manuscript and recommend reading our official comment for additional context.
>
> **Regarding the weaknesses:**
>
> **On real-world datasets (W1):** The lack of real large-component datasets with verifiable ground truth is precisely why synthetic benchmarks are essential. Beyond chemical applications, the framework applies to fixed-pattern signal unmixing at scale: hyperspectral unmixing with extensive material libraries, superposition feature decomposition in mechanistic interpretability, and physics signal analysis with large basis sets.
>
> These applications share two fundamental challenges: (1) existing MF-based methods cannot handle large candidate pools without exhaustive manual search over component numbers, and (2) **MF methods must re-optimize from scratch for each new sample**, whereas EB-gMCR's learned generative model can rapidly decompose new observations without retraining—critical for high-throughput analysis.
>
> Our validation strategy: Real datasets (Carbs N=3, NIR N=2) validate **correctness** (Fig. 3, Table 5); synthetic experiments validate **scalability** (Fig. 2, N>64). This approach is standard when ground truth at scale is unavailable.
>
> **On hyperparameter selection (W2):** We address each parameter:
>
> * **$\lambda$:** Dynamically adjusted by the algorithm (Eq. 13, Algorithm 1 line 14), not tuned. Scales automatically based on loss magnitude.
>
> * **$\lambda_{\text{se}}$ and $\lambda_{\text{amb}}$:** Set to $10^{-10}$ across all experiments. Convergence analysis (Appendix B, Lemma B.3) confirms small $\lambda_{\text{se}}$ ensures smooth optimization.
>
> * **Temperature $\tau$:** Key design is train/test gap. Training uses $\tau \geq 0.4$ for exploration; evaluation uses $\tau_{\text{eval}} = 0.01$ for near-deterministic predictions. This gap is the critical design choice.
>
> * **$R^2$ checkpoint bands:** User-defined quality targets based on expected noise (e.g., [0.97, 0.975) for high-noise, [0.99, 0.995) for low-noise), not algorithmic hyperparameters. Without ground truth, we cannot automatically determine which checkpoint best represents true decomposition.
>
> **Evidence of robustness:** Same settings work across N=32–256, 20–30dB noise, 4N–8N samples (Fig. 2, Tables 1–3).
>
> **On mathematical formalism (W3):** We systematically revised notation—explicit dimensions (e.g., $D \in \mathbb{R}^{M \times d}$) and clarified energy notation ($E$ for selection energy, $\varepsilon$ for measurement noise).
>
> **Regarding the questions:**
>
> **Q1 (Proof sketches):** Individual steps use standard techniques (sub-Gaussian concentration, PL-condition convergence). Our intent is verifying the multi-term loss (Eq. 13) converges end-to-end through two phases: support stabilization (Phase A) and fixed-support optimization (Phase B). Proof sketches highlight convergence pathway while keeping focus on framework development.
>
> **Q2 (Eq. 3):** Structural property following from Eq. 2 when each component's indicator is modeled independently. We revised Eq. 3 to make this explicit, noting it simplifies to $\mathbf{D}_{N+1}(\omega) \approx \mathbf{D}_N(\omega) + \mathbf{D}_1(\omega)$ for classical linear MCR.
>
> **Q3 (Eq. 5):** We clarified Eq. 5 in revised manuscript. Function $f_e$ uses Gumbel-softmax reparameterization. Approximation $\approx$ reflects temperature-controlled transition: training uses soft probabilities ($\tau \geq 0.4$) for gradients; evaluation uses $\tau_{\text{eval}} = 0.01$ for near-deterministic binary selection.
>
> **Q4 ($E_{f_e}$ in Eq. 6):** These refer to the same quantity—selection energy from EB-select module $f_e$. We unified notation in revised manuscript.
>
> **Q5 (Metrics - $R^2$ and EC):** We added "Evaluation Metrics" in Section 5:
>
> - **$R^2$:** Standard MCR metric for decomposition quality. Even under linear mixing, noise and unknown true forms mean perfect reconstruction is impossible. $R^2$ measures how well learned factorization reconstructs observed data—the only available ground truth. Values near 1 indicate effective decomposition.
>
> - **EC (Estimated Component number):** Counts components with selection probability exceeding $\tau_{\text{thres}} = 0.9999994$ (5σ confidence). Evaluates EB-gMCR's automatic component discovery—a core contribution. Baselines don't report EC because they require $N$ as input.
>
> **Additional clarification:** EB-gMCR's generative formulation enables inference on new samples without retraining (illustrated in our official comment). Traditional MCR must re-optimize entirely when new mixtures arrive. This reusability is key for high-throughput applications.
>
> We hope these clarifications address the reviewer's concerns.

---

### Official Review · Reviewer_Ljcq · 2025-11-01

**Soundness:** 2
**Presentation:** 1
**Contribution:** 3
**Rating:** 4
**Confidence:** 2

**Summary:**

This paper proposes an eneger-based method, EB-gMCR, to tackle multivariate curve resolution.
EB-gMCR automatically discovers the minimal set of latent components needed to reconstruct mixed signals by using a hard selection mechanism (EB-select) and additional regularizations to prune an initially oversized component pool during training. The approach shines when to hundreds of candidate components and integrates chemical priors naturally. Experiments on synthetic datasets (up to 256 components) and real spectral mixtures show that EB-gMCR accurately recovers component numbers and achieves strong reconstruction fidelity, outperforming classical MCR-ALS, NMF variants, and ICA particularly in high-component regimes.

**Strengths:**

- The proposed formulation is conceptually natural for the problem, and as a result the method exhibits favorable scaling compared to traditional matrix-factorization–based approaches.
- The framework is flexible, allowing domain knowledge to be incorporated.
- The empirical results are strong: the approach performs well in both synthetic benchmarks and real spectral datasets.

**Weaknesses:**

- The exposition is generally unclear. The core components of the method are spread across Section 4, with implementation details and conceptual justification interleaved, making it difficult to follow the full pipeline end-to-end. The presentation would benefit from consolidating the algorithmic steps (e.g., perhaps through a dedicated algorithm block or overview subsection) and then discussing each module in isolation in separate sections.
- In addition, there are several minor writing and notation issues, e.g. $\lambda_{\text{amb}}$ appears for the first time in Eq. 13 without a clear linkage to the ambiguity regularizer in Eq. 9, and $X_o$ is not introduced prior to line 201. A few other are listed below, but they are frequent enough that a careful pass is warranted.
- One of the main benefits of the method is it's computation efficiency as number of components grow. However, the experiments on real datasets involve only 2-3 components. Could the authors demonstrate performance of EB-gMCR in a real, large-component setting?

**Questions:**

- In line 240, why $\tau = 0.9999994$?
- In line 302, what is PL?
- In line 202, do you mean $X_o$, as opposed to $X_0$
- Could the authors provide wall-time comparison between EB-gMCR and baselines, particularly when component count is high.

---

> ### Author Response · Authors · 2025-11-20
> **Response to Reviewer Ljcq.**
>
> We sincerely appreciate the reviewer's careful attention to mathematical presentation clarity. We have thoroughly revised the paper and reformatted the mathematical equations. The main revisions focus on: (1) providing clear definitions for all notation in equations, and (2) explicitly denoting matrix dimensions. We believe the revised main text provides significantly improved readability and logical flow.
>
> **Regarding the weaknesses:**
>
> **On exposition clarity (W1):** We have restructured Section 4 to improve separation of concerns. Algorithm implementation details now reside in Section 4.4 (Checkpointing Solver), while Section 4.5 focuses exclusively on convergence analysis. We believe this separation provides readers with a clearer conceptual framework.
>
> **On notation issues (W2):** We have systematically corrected all notation inconsistencies:
> - $\lambda_{\text{amb}}$ is now introduced when first discussing the HSIC regularizer (around Eq. 9, before Eq. 13)
> - All instances of $X_0$ have been corrected to $X_o$ (where "o" denotes "observed," corresponding to collected data D)
> - PL is now explicitly written as Polyak-Łojasiewicz at first use
>
> **On real large-component experiments (W3):** We appreciate this question. The lack of real large-component datasets with verifiable ground truth is precisely why we developed synthetic benchmarks—they enable rigorous validation that real data cannot provide.
>
> **The framework enables future applications.** While our chemical motivation is clear (library-based searches in mass spectrometry and metabolomics involve hundreds to thousands of candidate components, the framework applies broadly to **any fixed-pattern signal unmixing problem at scale**. Beyond chemistry, potential applications include:
> - Hyperspectral unmixing with extensive material spectral libraries
> - Superposition feature decomposition in mechanistic interpretability (identifying which learned features activate in neural network representations)
> - Physics signal analysis with large basis sets (e.g., decomposing detector signals into particle signatures)
>
> These applications share two fundamental bottlenecks that existing MF-based methods face: (1) **exhaustive manual search over component numbers for large candidate pools**, and (2) **inability to reuse learned decomposition patterns**—traditional MCR must re-optimize the entire factorization when new samples arrive. EB-gMCR's generative formulation addresses both: automatic component discovery through energy-based selection, and fast inference on new samples via the learned generative model.
>
> **Our validation strategy:** Real datasets (Carbs N=3, NIR N=2) validate **correctness**—EB-gMCR recovers the right component count and achieves competitive decomposability (Fig. 3, Table 5). Synthetic experiments validate **scalability**—EB-gMCR maintains performance where baselines fail (Fig. 2, N>64). This two-pronged approach is standard in ML when ground truth at scale is unavailable.
>
> **Future work:** We are actively exploring applications in mass spectral library search (500+ candidate compounds) and other domains. However, such datasets typically lack ground truth for quantitative validation, making synthetic benchmarks essential for controlled method development and comparison.
>
> The framework's value lies in enabling a class of problems that were previously intractable. We view this paper as establishing the methodology and demonstrating its feasibility at scale.
>
> **Regarding the specific questions:**
>
> **Q1 (Line 240, $\tau_{\text{thres}} = 0.9999994$):** This value corresponds to the 5σ confidence level commonly used in physics experiments to establish signal existence (equivalent to p < 3×10⁻⁷). We provide this justification in Lines 260-261 of the revised manuscript.
>
> **Q2 (Line 302, PL):** PL now explicitly states "Polyak-Łojasiewicz" at first mention.
>
> **Q3 (Line 202, $X_0$ vs $X_o$):** We apologize for this inconsistency. All instances have been corrected to $X_o$ throughout the manuscript.
>
> **Q4 (Wall-time comparison):** We have added Table 1, which provides wall-clock time comparisons between EB-gMCR and baselines at multiple search intervals (8, 16, 32). We acknowledge this comparison is inherently asymmetric (traditional MCR methods search upward from small N, while EB-gMCR initializes with N=1024 and prunes). However, Table 1 demonstrates the key practical advantage: EB-gMCR eliminates the need for manual N specification and exhaustive search, which becomes prohibitively expensive at scale.
>
> We hope these revisions and clarifications address the reviewer's concerns. We are happy to provide additional details if needed.

---

### Author Response · Authors · 2025-11-20
**(Author Comment) Foundational Clarifications on gMCR Framework.**

Thank you to all reviewers for your thoughtful feedback. Before addressing individual concerns, we clarify the core problem to bridge potential understanding gaps.

We provide a toy example to illustrate what MCR is. In short, MCR decomposes observed superposition signals into underlying components and their concentrations. Consider this simple example:

**Given observations:**
$$
\\begin{aligned}
\\text{Sample 1:} \\quad &[1, 1, 0, 0] = 1 \\times C_1 \\\\
\\text{Sample 2:} \\quad &[0, 0, 1, 2] = 1 \\times C_2 \\\\
\\text{Sample 3:} \\quad &[1, 1, 1, 2] = 1 \\times C_1 + 1 \\times C_2
\\end{aligned}
$$

**The decomposition is:** $C_1 = [1,1,0,0]$, $C_2 = [0,0,1,2]$ (not normalized for visual clarity) with concentrations $[[1,0], [0,1], [1,1]]$.

**Why is this "generative"?** Each observation is generated by: mixture $= \\sum (\\text{concentration} \\times \\text{component})$. Real-world signals with fixed patterns can be combined in more complex non-linear ways, so we use an aggregation function $\\Phi$ instead of simply assuming linear addition. This compositional structure is the natural description of how superposition signals form. We model this generation process to perform decomposition (conditional inference), not to synthesize new mixtures.

**Traditional MCR's main weakness is lack of scalability.** This manifests in two ways: scaling to large datasets, and decomposing new samples efficiently. Traditional methods initialize fixed-size matrices: concentration $C \\in \\mathbb{R}^{M \\times N}$ and components $S \\in \\mathbb{R}^{N \\times d}$, where $N$, $M$, $d$ are predefined component number, collected sample size, and data dimension. As the toy example shows, we can easily collect more data from the same signal source and expand the dataset:

$$
\\begin{aligned}
\\text{Sample 1:} \\quad &[1, 1, 0, 0] = 1 \\times C_1 \\\\
\\text{Sample 2:} \\quad &[0, 0, 1, 2] = 1 \\times C_2 \\\\
\\text{Sample 3:} \\quad &[1, 1, 1, 2] = 1 \\times C_1 + 1 \\times C_2 \\\\
\\text{Sample 4:} \\quad &[2, 2, 0, 0] = 2 \\times C_1 \\\\
\\text{Sample 5:} \\quad &[0, 0, 2, 4] = 2 \\times C_2
\\end{aligned}
$$

Intuitively, $C_1$ and $C_2$ remain the main components, but traditional MCR must recompute everything when any new sample is added. This toy example uses clean signals, but recomputation amplifies noise effects, causing numerical instability.

**For scaling to large component numbers or dataset sizes:** Specifying $N$ before running the algorithm means we must search over candidate $N$ values and rerun the algorithm for each one. This becomes prohibitively expensive when $N$ and $M$ are large. Moreover, we found that traditional MCR methods cannot maintain effective decomposability even when given the correct $N$. They fail to preserve reconstruction quality when $N$ exceeds several hundred. While we tested up to $N=256$, EB-gMCR maintains better decomposability as $N$ increases (Fig. 2f and 2h: near-ground-truth component counts at N=256).

**Regarding the convergence proof:** We aim to justify why and how the complex loss function (Eq. 13) can reach an endpoint through end-to-end training. The proof does not claim to prove why this framework and loss function work for signal unmixing—the framework is heuristic and the design comes from physical intuition. We hope this clarification helps reviewers better understand the problem we address and the roadmap this paper provides for solving signal unmixing challenges.

We address specific reviewer questions in individual responses below.

---

### Meta-Review · Area_Chair_kARa · 2026-01-16

**Summary:**

**Reviewer Ljcq:** Presentation unclear (W1). Writing and notation issues (W2). Experiments were done on datasets with a few components (W3). Wall-clock comparison missing (Q4).

**Reviewer zgrb:** Only relatively simple datasets were tested (W1). There are several hyperparameters, and how to select them and how sensitive they are are not thoroughly discussed (W2). Mathematical presentation is difficult to follow (W3). Only proof sketches are provided (Q1).

**Reviewer NBQX:** A combination of heuristics without motivation provided for choices (W1). Some instances of notation that are not defined (W2). Comparison in computation time is missing (W3).

**Reviewer TaiB:** Novelty is limited (W1). Claimed plug-in cross-domain adaptation is not backed up (W2, Q5). Experiments are too simple (W3). No generative metrics are reported (W4). Hyperparameters would need manual tuning (W5, Q1). Unclear description (W6). Extensibility to nonlinear tasks (Q2).

**Additional points:**
- Line 242: To further (fasten → accelerate?) convergence,
- Line 792: $R$ (i)s any
- Line 794: in (below → the following) convergence analysis
- Ambiguous definition of symbols.
  - $\tau$: Its first appearance of this "temperature" parameter in this paper is in line 231, whereas how it is implemented in the function $f_e$ is not described in the manuscript.
  - $e_t^{(i)}$: Although its first appearance is equation (6), there would be no definition provided so far. I guess that the selection energy matrix $E$ is determined in terms of the output of the function $f_e$, but how it is determined, as well as its relation with $e_t^{(i)}$ and $E_{f_e}$ (cf. NBQX), is not described.
  - $\delta$: Upto equation (5), $\delta$ is a random quantity. (I guess that it is an $N$-dimensional random binary vector, $\delta\in\\\{0,1\\\}^N$.) On the other hand, in equation (8) $\delta$ seems to be a deterministic quantity.
  - $S^\*$: In line 803, $S^\*$ is multiplied with $c_i^\*$, whereas in line 813, the same symbol $S^\*$ denotes a set.
- Assumption B3: This is the conventional Lipschitz continuity of $\Delta_j$ with the Lipschitz constant $L_\Delta$, and therefore it should be stated as such.
- Corollaries, Lemmas, Theorems: The statements in most of them are too lengthy, sometimes with extra definitions inside them. Take Corollary B.1 as an example. It contains the definitions of "Phase A/B", which should be put outside the corollary. I do not understand what "the variance Lemma B.5 / Theorem B.6" means. I think that, overall, the quality of writing in the mathematical formulation and the theoretical developments does not reach the level of common expectation in our research field.

**Reviewer Concerns:**

**Lack of concrete proofs (zgrb):** The author responded to the review comment just by writing "Proof sketches highlight convergence pathway while keeping focus on framework decelopment," and yet I cannot understand most of the mathematical development in this paper.

**Tested on simple datasets only (Ljcq, zgrb, TaiB):** Although I would agree that ground-truth information should be needed to evaluate performance, in real chemical analysis it would often be the case that decomposition is not the ultimate goal, so that one may be able to choose an appropriate downstream task and to see how the proposal would improve performance of the downstream task for the purpose of evaluating efficiency of the proposal in practical scenarios.

**Computation time (Ljcq, NBQX):** Table 1 has been added to show comparison in wall-clock time.

**Reviewer Scores:**

All the review scores were on the negative side of the acceptance threshold, and I see no reasons for the reviewers to change their evaluations positively after the discussion period. Furthermore, as mentioned above, the mathematical formulation and the theoretical developments in this paper are not well organized.

---

### Decision · Program_Chairs · 2026-01-26

Reject